# How Capable Can a Transformer Become?
# A Study on Synthetic, Interpretable Tasks

## Abstract

Transformers trained on huge text corpora exhibit a remarkable set of capabilities, e.g., performing simple logical operations. Given the inherent compositional nature of language, one can expect the model to learn to compose these capabilities, potentially yielding a *combinatorial explosion* of what operations it can perform on an input. Motivated by the above, we aim to assess in this paper "how capable can a transformer become?". Specifically, we train autoregressive Transformer models on a data-generating process that involves compositions of a set of well-defined monolithic capabilities. Through a series of extensive and systematic experiments on this data-generating process, we show that: (1) Autoregressive Transformers can learn compositional structures from the training data and generalize to exponentially or even combinatorially many functions; (2) composing functions by generating intermediate outputs is more effective at generalizing to unseen compositions, compared to generating no intermediate outputs; (3) the training data has a significant impact on the model's ability to compose unseen combinations of functions; and (4) the attention layers in the latter half of the model are critical to compositionality.

## 1 Introduction

Large scale Transformers pretrained on huge text corpora have revolutionized machine learning in recent years (Radford et al., 2018; 2019; Brown et al., 2020; Sanh et al., 2021; Wei et al., 2021; Thoppilan et al., 2022; Touvron et al., 2023). Due to an ever-increasing interest in adopting these models in our daily lives, evaluating and predicting their capabilities has become increasingly important (Bommasani et al., 2021; Ganguli et al., 2022; Shevlane et al., 2023; Rae et al., 2021; Hoffmann et al., 2022; Tay et al., 2022; Henighan et al., 2020; Hernandez et al., 2021; Sharma & Kaplan, 2020). Motivated by this, several recent works have performed extensive empirical analyses to better understand the possibilities and limitations of using these models in practical tasks of interest. For example, such works show large language models (LLMs) can generate coherent text completions based on a provided context, perform code generation and debugging, use online APIs and tools in an automated manner, and even solve multimodal problems such as image captioning (Wei et al., 2022a; Bubeck et al., 2023; Austin et al., 2021; Chen et al., 2021; Lee et al., 2023; Liang et al., 2022; Qin et al., 2023; Liu et al., 2023; Merullo et al., 2022; Suzgun et al., 2022; Srivastava et al., 2022). While this benchmarking of pretrained models is extremely valuable, it often focuses on evaluating rather "primitive" capabilities; for example, the ability to identify whether a given passage of text is biased or toxic (Gehman et al., 2020; Liang et al., 2022). However, given the compositional nature of data these models are trained on (e.g., language), it is possible that a model learns to *compose* its primitive capabilities, hence yielding abilities to perform tasks that we never explicitly trained it for. This can lead to an underestimation of the capabilities the model possesses and what tasks we can expect it to perform; vice versa, if the model does not learn to compose, we can be certain that benchmarking for primitive capabilities is sufficient to characterize the model.

Motivated by the above, our goal in this work is to analyze if a Transformer trained on a compositional data-generating process, without any special modifications to the usual training pipeline, can learn *both* relevant primitive capabilities and an ability to compose those capabilities. As shown by Bubeck et al. (2023), LLMs have already started to exhibit "sparks" of such compositional capabilities, such as generating text that merges content of varying styles or evaluating mathematical expressions through the application of a sequence of functions (Fig. 1). However, due to their black-

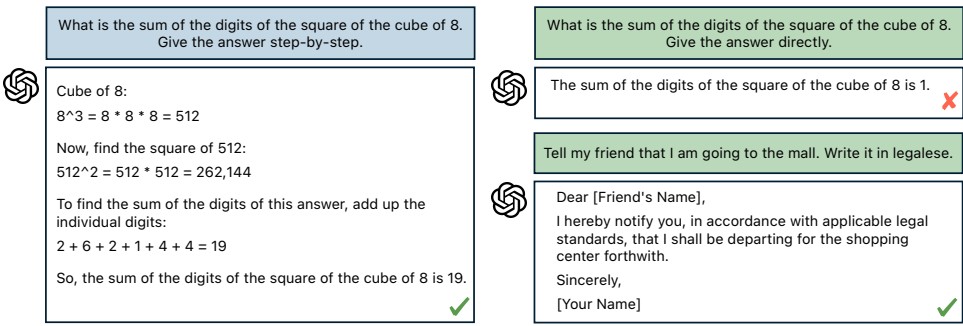

Figure 1: **Signatures of compositionality.** ChatGPT (Bubeck et al., 2023) correctly responds to prompts that require composition of primitive arithmetic capabilities (sum, cube, square)—we argue these prompts are unlikely to be in the training data. However, the model does not always compose reliably (top-right panel). This motivates us to study the extent to which a Transformer can learn to compose its capabilities by mere pretraining on a compositional domain.

box nature, it is unclear if an LLM actually learns to compose capabilities or merely memorizes relevant samples from its training data. Moreover, while interacting with an LLM, it can be difficult to guarantee that we are utilizing a prompt that will appropriately guide the model to use the capabilities we desire, let alone compose them. Correspondingly, we may end up claiming the model lacks a certain capability, when in fact we may not be utilizing the appropriate context for eliciting it (Suzgun et al., 2022; Reynolds & McDonell, 2021; Lu et al., 2023; Wei et al., 2022b).

To circumvent challenges faced with LLMs pretrained on real world data and focus on our specific motivation, *"can a Transformer trained on compositional data learn to compose its capabilities"*, we choose to limit the purview of this work to a well-defined synthetic domain. This is similar in spirit to several recent works that utilize synthetic datasets generated using objects like first-order logic machines, context-free grammars, linear regressors, modular arithmetic, and even board games to establish and understand phenomenology of modern neural networks (Liu et al., 2022; Allen-Zhu & Li, 2023c;a;b; Garg et al., 2022; Li et al., 2023b; Saparov & He, 2022; Chan et al., 2022; Bhattamishra et al., 2020; Zhou et al., 2023; Nanda et al., 2023a;b; Li et al., 2023a; Lubana et al., 2023; Jones, 2021). The goal of such works, including ours, is to develop interpretable demonstrations and mechanistic hypotheses that enable a characterization of the target phenomenology in a controlled setting. Accordingly, we emphasize we do not intend to develop novel protocols for improving Transformers' ability to compositionally generalize, but rather to demonstrate and understand what drives its existence in the first place. Overall, we make the following contributions in this work:

(1) **A minimal synthetic setup for characterizing Transformers' ability to compose.** We propose a minimal setup involving compositions of predefined functions $\mathcal{F}$ (bijections and permutations) that operate on a string of arbitrary tokens (Sec. 3). Motivated by instruction induction and tuning in LLMs (Honovich et al., 2022; Wei et al., 2021), we instantiate a notion of "task tokens" which specify what functions are to be applied to the input string. This helps us avoid any ambiguity in task-specification (Suzgun et al., 2022; Si et al., 2023).

(2) **Transformers show explosion of capabilities.** We characterize the ability of a Transformer trained autoregressively on our proposed setup to compositionally generalize, i.e., to apply a composition of specific functions chosen from $\mathcal{F}$ to an input string. As we show, the model can generalize to exponentially or even combinatorially many functions (Sec. 4.1)—these functions are entirely "out-of-distribution", i.e., the model never sees them in its training data and hence was not explicitly trained to learn them. The crucial component here is the use of stepwise inference, i.e., allowing the model to recursively process its intermediate outputs (Sec. 4.3).

(3) **Characterizing limitations and mechanisms of compositionality in a Transformer.** We formalize a notion of "distance" between the functions seen by the model during pretraining and the ones it is evaluated on, hence enabling a precise characterization of situations wherein the model struggles to compose (Sec. 4.2). As we show, the training data non-trivially determines whether the Transformer generalizes to an exponential (which we call in-order generalization) or combinatorial (which we call out-of-order generalization) set of functions. Furthermore, by using the popular linear probing protocol used for understanding Transformer internals (Tenney et al., 2019; Li et al., 2023a), we show Attention layers in the latter half of the model play a crucial role in enabling compositional generalization in a Transformer (Sec. 4.4).

## 2 RELATED WORK

**Capabilities in a Transformer.**   Transformers pretrained on large-scale, web-crawled datasets have been shown to exhibit a slew of interesting capabilities, such as primitive arithmetic, question answering, commonsense knowledge reasoning, stylistic transformation of a piece of text, and even multimodal reasoning (Radford et al., 2018; 2019; Brown et al., 2020; Bubeck et al., 2023; Wei et al., 2022a; 2021; Rae et al., 2021; Chowdhery et al., 2022; Austin et al., 2021; Chen et al., 2021; Bommasani et al., 2021). However, this generality can come at the cost of a model also learning capabilities that are undesirable (Bommasani et al., 2021; Tamkin et al., 2021; Chan et al., 2023), e.g., producing sensitive, biased, or toxic outputs (Weidinger et al., 2021; McGuffie & Newhouse, 2020; Garrido-Muñoz et al., 2021; Lin et al., 2021; Jiang et al., 2021; Abid et al., 2021; Parrish et al., 2021; Xu et al., 2021; Huang et al., 2019; Sheng et al., 2019; Gehman et al., 2020; Xu et al., 2020; Tamkin et al., 2021). This has motivated several works focused on understanding capabilities of a pretrained model, including (i) *predicting* capabilities of a *future* model, e.g., via fitting power laws to data/model scaling results (Rae et al., 2021; Hoffmann et al., 2022; Hernandez et al., 2021; Sharma & Kaplan, 2020) and (ii) *eliciting* capabilities of a *given* model, e.g., via identification of appropriate prompts or via step-wise inference protocols such as chain-of-thought, to understand what tasks a the model can be reliably used for (Liang et al., 2022; Suzgun et al., 2022; Lee et al., 2023). However, we argue that by measuring a model's performance on benchmark tasks to identify or predict the existence of a specific set of capabilities is bound to be insufficient for characterizing what tasks it can perform: given the compositional nature of data that modern neural networks are trained on, it is possible that they learn how to *compose* capabilities, hence learning how to perform several more tasks than we explicitly train or evaluate them on.

**Compositionality in neural networks.**   The ability to compositionally reason has been touted as a cornerstone of human intelligence (Fodor & Lepore, 2002; Fodor & Pylyshyn, 1988; Fodor, 1975; Schulz et al., 2016). Accordingly, several works have studied the ability of a neural network to compositionally generalize, generally demonstrating a negative result, and correspondingly developing explicit strategies that help improve the model's ability to generalize (Liška et al., 2018; Hupkes et al., 2018; Lake & Baroni, 2018; Csordás et al., 2021b;a; 2022; Ontanón et al., 2021; Lepori et al., 2023; Lewis et al., 2022; Yun et al., 2022; Okawa et al., 2023; Hosseini et al., 2022). Our work differs from prior literature in several manners. (i) We do not intend to develop protocols for improving compositional generalization in a Transformer; instead, our goal is to show that mere autoregressive training on strings generated using a compositional data-generating process can yield a Transformer that can compose its capabilities and perform tasks it was never explicitly trained for. To this end, we define a synthetic task that allows for perfect task specification and hence helps avoid ambiguity due to prompt misspecification. While similar to the compositional table lookup task used in prior work (Liška et al., 2018; Csordás et al., 2022), our task involves a much larger set of capabilities to train and test for (3125 or 4 million, depending on the setup, compared to 128 capabilities in prior work). (ii) We aim to understand the extent of compositional generalization in a Transformer trained on our proposed domain, i.e., what kind of compositions does the model fail to perform and when. We define a framework to precisely characterize these failures modes and use the popular linear probing protocol for understanding model internals to show the critical role of attention layers in enabling compositionality (Li et al., 2023a). (iii) Finally, we analyze the impact of step-wise inference protocols, wherein intermediate outputs generated by the model are recursively passed to it as inputs, and which has been used for solving several challenging benchmark tasks recently (Suzgun et al., 2022; Wei et al., 2022b). While a few prior works have studied, in a similar spirit as ours, whether a Transformer can learn to compositionally generalize (Csordás et al., 2021a; Ontanón et al., 2021), we emphasize these works focus on compositionality via a singular forward pass, i.e., the model is not allowed to recursively process its inputs. We find the use of intermediate outputs significantly simplifies the problem and, given its popularity in practical scenarios, our results serve as a demonstration that inference protocols that allow Transformers to recursively refine their outputs can lead to a wide range of capabilities, especially ones that we never explicitly train the model for.

## 3 FORMALIZING CAPABILITIES AND COMPOSITIONS

As noted by Hupkes et al. (2020), despite extensive work exploring compositionality in neural networks, the term is often used for several related concepts. To avoid ambiguity, we thus present a

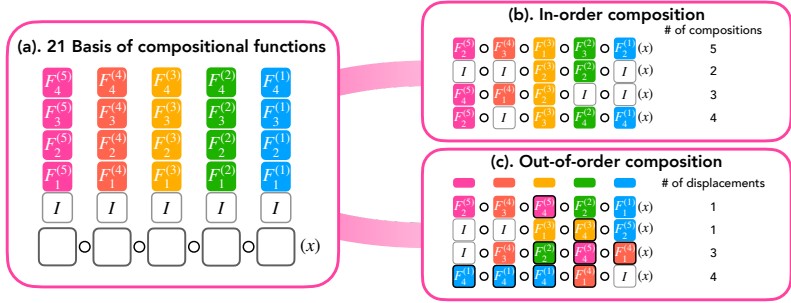

Figure 2: **Data generating process for in-order and out-of-order compositions.** (a) Each of the $L = 5$ positions is associated with $N = 4$ functions $f_i^{[l]}$, in addition to an identity function, resulting in a total of $5 \times 4 + 1 = 21$ basis functions for composition. (b) The in-order compositions select functions within the same position while (c) out-of-order compositions allow for selecting functions across positions. Each position also includes the identity function since it allows us to compute compositions of fewer than 5 functions. In the examples presented in (c), displaced functions are surrounded by a black line, and we then count the number of displaced functions.

definition of a "compositional model" that captures our intended notion and, correspondingly, describe the data-generating process used in this work to understand Transformers' ability to compose.

Let $\mathcal{F}$ denote a set of predefined automorphisms, i.e., any given function $F$ from the set defines a map between points from its input space to the same space. This is motivated by the fact that the input and output domain of a language model are generally the same. We define an **input** $x$ as a combination of two strings $[x_f, x_d]$, where $x_f \in X_f^L$ is a sequence of $L$ tokens that specify a series of $L$ functions from $\mathcal{F}$ that are to be sequentially applied to a $k$ token sequence specified by $x_d \in X_d^K$, where $|X_d| = V$. We refer to $x_f$ as **task tokens** and to $x_d$ as **data tokens**. For example, let $x_{F_i}$ be the identifier that denotes that function $F_i$ is to be applied to the data tokens and $x_{d_k}$ denote the $k^{\text{th}}$ token from the vocabulary $X_d$. Assume $L = 2$ and $k = 1$ and define a sample $x = [x_{F_1}, x_{F_2}, x_{d_1}]$. Then, a model $M : X_f^L \times X_d^K \mapsto X_d^K$ that takes $x$ as input is expected to produce the output $F_2 \circ F_1 (x_{d_1})$. We use $[L]$ to denote the ordered set $\{1, 2, \ldots, L\}$.

A **capability** is defined in our setup as the ability of a model to accurately represent a function $F \in \mathcal{F}$. We emphasize that we do not expect pretrained models in practice to perfectly implement an arbitrary function; however, this idealized definition affords us precision by allowing us to use accuracy over a random set of inputs to claim a model possesses a certain capability. Based on this definition, we intend to understand the set of capabilities—or the set of functions—that a Transformer can implement by composing them. We formalize this as follows.

**Definition 1 (Compositionality.)** *We say a model $M(.)$ compositionally generalizes if, for any subset of functions $F_i \in \mathcal{F}$, where $i \in [L]$, $M\left([x_{F_1}, x_{F_2}, \cdots x_{F_L}, x_d]\right) = F_L \circ \cdots \circ F_2 \circ F_1 (x_d)$.*

In practical scenarios, we would not expect the pretraining data to present a capability in all possible scenarios that it can be used in. For example, simple arithmetic tasks like multiplication are often only seen in the context of numbers 1–3 digits in web-crawled data (Razeghi et al., 2022), which leads to an inability of the model to perform multiplication in higher order numbers. To model this in our setup, we create a spurious correlation between a subset of the functions from $\mathcal{F}$ and the position of their identifiers in the task tokens $x_f$. Specifically, we define $\mathcal{F}^{(l)} \subset \mathcal{F}$ as the set of functions that are allowed at the **position** $l$ in the task tokens $x_f$ of a datapoint $x$. We let $|\mathcal{F}^{(l)}| = N$ for all locations $l$, i.e., $\mathcal{F}$ is partitioned into equally sized subsets and $|\mathcal{F}| = N \times L$. The notation $F_i^{(l)}$, where $i \in [N]$ and $l \in [L]$, is used to denote the $i^{\text{th}}$ possible function at position $l$. Based on the above, we define two ways to compose $L$ functions: **in-order** and **out-of-order** (see Fig. 2).

**Definition 2 (In-order vs. out-of-order Compositions.)** *Consider the composition $\widetilde{F} = F^{(l_1)} \circ \cdots \circ F^{(l_2)} \circ F^{(l_L)} (.)$, where $l_i \in [L]$. Denote the ordered set $\{l_1, l_2, \ldots, l_L\}$ as $\mathtt{order}(\widetilde{F})$. If $\mathtt{order}(\widetilde{F})$ equals the set $[L]$, we say $\widetilde{F}$ is an in-order composition; else, we say it is out-of-order.*

Consider a model $M$ that perfectly encodes all $N \times L$ functions from the set $\mathcal{F}$. If the model can generalize to *in-order* compositions of these functions, then its set of capabilities will in fact grow

to exponentially many functions—$N^L$ of them to be precise. Further, the ability to compose *out-of-order* can increase this set combinatorially, i.e., proportional to $(N \times L)^L$, growing even more quickly compared to the set of in-order compositions. Such an "explosion of capabilities" would imply perfect knowledge of what all tasks a pretrained model can perform is difficult to characterize or predict, especially since the pretraining data used for training a model is generally unknown and hence it is hard to characterize even what "primitive" capabilities the model possesses. In our experiments, we find that while Transformers can generalize to both in-order and out-of-order compositions, the pretraining dataset for enabling out-of-order generalization must exhibit sufficient (albeit not huge) diversity. To empirically characterize this and discuss the failure modes on out-of-order compositions, we find it useful to define the following notion of **displacement**.

**Definition 3 (Displacement.)** *Let $D(s, s')$ denote the hamming distance between two ordered sets $s$ and $s'$. Then, the displacement of a composition $\widetilde{F}$ is defined as $D(\texttt{order}(\widetilde{F}), [L])$.*

## 3.1 EXPERIMENTAL SETUP AND DATA-GENERATING PROCESS

Having defined our notion of compositionality of capabilities in a pretrained model, we now briefly discuss the experimental setup used in this work (see App. A for details). Specifically, our data-generating process yields inputs consisting of a sequence of 6 **data tokens**, $x_d \in X_d^6$, where each token is drawn from a vocabulary of size $|X_d| = 10$. Each of the 6 elements are drawn uniformly at random, with replacement, from $X_d$. We consider **two families of functions** defined over these data tokens: bijections and permutations (see Figure 10) Specifically, the set $\mathcal{F}_b$ (which we refer to as bijections) consists of all functions that apply a bijection on each of the 6 tokens in an element-wise manner. The number of such functions is the number of bijections on a single token: there are 10! such functions when $|X_d| = 10$. The second set is $\mathcal{F}_p$, which is the set of all permutations of 6 elements ($|\mathcal{F}_p| = 6!$). The rationale for selecting these function families is that both $\mathcal{F}_b$ and $\mathcal{F}_p$ are groups with function composition as the group operator. As a result, the compositions of two functions will also be an element in the group.

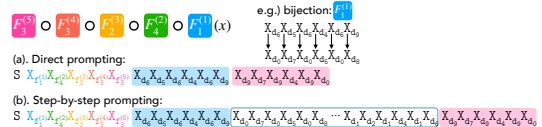

Figure 3: **Direct v.s. Step-by-step prompts.** The task (rainbow) and data (blue) tokens can be completed in two ways. They are followed by the intermediate outputs of the composition in the step-by-step format **(a)** or directly by the final result of the series of compositions in the direct format **(b)**.

We also control the set of **task tokens** seen during training. For example, we can choose to partition the set of functions into different subsets and only include in-order compositions in the training data. We define two subsets of of the function class $\mathcal{F}_b$: **random** and **21 base** (see Appendix A.2). The set **random** contains a random set of compositions of functions from the set of all possible in-order compositions. The set of functions **21 base** considers compositions of 5 functions, where at least 4 are the identity function. Each of the 5 positions have 4 choices for the function which totals to 21 functions if we include the identity function. This set helps us assess whether mere learning of "primitive" capabilities is sufficient to yield compositionality in a model. We consider two formats for representing a sample (see Fig. 3). Both formats start with task tokens $x_f$, that specify the sequence of functions to compose, followed by the data tokens $x_d$. The **direct prompt** format follows this with the final output of the function composition, while the **step-by-step prompt** format follows this with all intermediate outputs of the function composition. We generate 100,000 samples using the process above for a given prompt format (step-by-step or direct) and with restrictions on the task tokens (in-order, out-of-order, **21 base**, **random**). The model is then trained on this data (see Appendix A) in an autoregressive manner using the cross-entropy loss. After training, we evaluate whether the model possesses a capability corresponding to a set of composition of functions (depends on the experiment) by computing the accuracy of the model completion on 1000 different data tokens. The accuracy of a completion is the average accuracy over the last 6 tokens.

## 4 RESULTS

In this section, we systematically investigate the capabilities of an autoregressive Transformer trained on synthetic tasks with compositional structure. Broadly, we would like to understand how

this structure in the data manifests in the network. We answer the following questions: (1) Do Transformers generalize to functions not present in the training data and to what extent do they exhibit in-order and out-of-order generalization? (2) How do properties of the training data influence in-order and out-of-order generalization? (3) Is there a difference between direct and step-by-step composition? (4) Do Transformers first learn to compose fewer functions before learning to compose many of them? (5) What is the role of attention and does it help a Transformer compose different functions? Additional results are presented in Appendix B.

## 4.1 COMBINATORIAL EXPLOSION AND EXPONENTIAL GROWTH IN CAPABILITIES

Do Transformers only generalize to functions present in the training data or do they reflect compositional structure present in data? In Fig. 4, we train on data consisting of a small subset of in-order compositions of functions from the set of bijections $\mathcal{F}_b$, in the step-by-step prompt format. We consider the composition of 5 functions in both Figures 4a and 4b. Each position of the composition can be one of 4 choices, with the 4 choices at different positions being different in Fig. 4a and the same in Fig. 4b. In addition, any position can also be selected to be identity.

**We find that a Transformer can capture the compositional structure in data and generalize to an exponential and combinatorial set of functions in Fig. 4a, 4b, despite being trained on an *extremely small* subset of function compositions.** For example, a Transformer trained on just 30-100 compositions of functions generalizes to 3125 unseen compositions of these functions almost perfectly. This could explain why language models show signatures of compositionality. In contrast, we note LSTMs fail to compositionally generalize in this same setup (Appendix B.2), while Transformers with different numbers of layers and attention heads show compositional generalization (Appendix B.1). This indicates that the **inductive bias of the architecture contributes to compositional generalization and any autoregressive model is not guaranteed to succeed.** We also observe that **21 base**—which serves as a null model that only trains on the monolithic capabilities (or functions)—does not compositionally generalize. In summary, compositional generalization occurs with the step-by-step prompt format, but also requires the right architecture and training data.

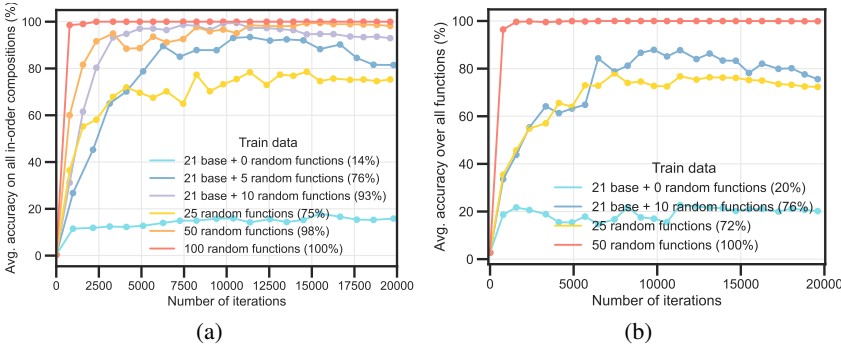

(a)                                           (b)

Figure 4: **Transformers can generalize to an exponential (a) or combinatorial (b) number of new functions.** We plot the accuracy averaged over all compositions of 5 bijections, where each position of composition has 4+1 choices, with one of them being the identity function. Each curve corresponds to training data generated by a different subset of functions and the model is trained using the step-by-step prompt format. **(a)** The choice of 5 functions are different at different positions of composition—there are 21 different functions (1 identity) which can be composed (in-order) in 3125 different ways. **(b)** The choice of 5 functions are identical across all 4 positions of the composition which means there are 3125 different ways to compose them; only 1365 of them are unique. Both figures are evidence that one can train on a small number of compositions of functions (around 31-100) and generalize to exponentially (a) and combinatorially (b) many functions that would be considered "out-of-distribution".

## 4.2 IN-ORDER VS. OUT-OF-ORDER GENERALIZATION

How do biases in the training data influence a Transformer's ability to compose? Are Transformers capable of in-order and out-of-order generalization and does it depend on the nature of training data?

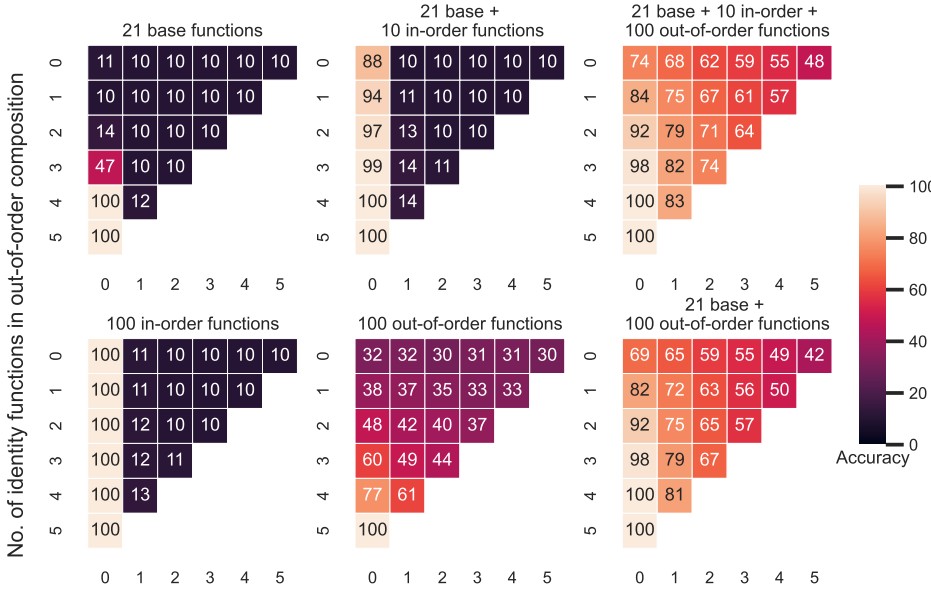

Figure 5: **The training data determines if a Transformer generalizes to an exponential (in-order generalization) or combinatorial (out-of-order generalization) number of functions.** Each subplot uses a different subset of functions (from $\mathcal{F}_b$) to generate the training data and we evaluate them on combinatorial set of functions generated from 20+1 functions (one of them being identity). The x-axis varies the number of displacements and the y-axis varies the number of compositions—equivalently the number of functions that are not identity. We make the following observations: (1) A Transformer trained on just 31 functions (top-middle) generalize to nearly exponentially many or 3125 compositions of functions. (2) All the above configurations do not generalize perfectly to the entire combinatorial set. They however partially generalize to nearly 4 million compositions of functions. The generalization is worse if we increase the number of compositions or displacements (see Fig. 2 for pictorial description of displacements).

For the functions in Fig. 4a, the number of in-order compositions is $5^5 = 3125$ and the number of out-of-order compositions is a whopping $(21)^5 = 4084101$; essentially all of these functions are different from the ones seen in the training data. Like in Sec. 4.1, we only consider Transformers trained with the step-by-step prompt format on functions from the set of bijections $\mathcal{F}_b$. In Fig. 5, we consider the training data to have functions from **21 base**, some in-order and some out-of-order compositions. We fail to see in-order or out-of-order generalization unless the data also includes in-order or out-of-order compositions respectively. **However, a small number of in-order (10 of them) or out-of-order compositions (100 of them) in the training data is enough for in-order generalization and limited out-of-order generalization.** All scenarios in Fig. 5 do not fully generalize to out-of-order compositions. This indicates that out-of-order compositions may require a lot more data compared to in-order compositions.

## 4.3 DIRECT VS. STEP-BY-STEP COMPOSITIONS

Both Sec. 4.1, 4.2 use step-by-step compositions, but do these results also hold for direct prompting? Fig. 6 (Left) and Fig. 15 answer this in the negative. Specifically, in Fig. 6 (Left), we consider a setup identical to Fig. 4a and train on a different number of **random** functions. **Transformers fail to generalize to new in-order compositions with direct prompting when we consider compositions of bijections from $\mathcal{F}_b$.** We observe this failure even if we train of 2000 of the 3125 possible in-order compositions of functions, i.e., even if the data has high diversity. In contrast, in Fig. 4a, mere 100 compositions in the step-by-step format suffices to generalize to all possible in-order compositions.

**On the other hand, we see in-order generalization if a Transformer is trained on a composition of a a permutation function from $\mathcal{F}_p$ and a bijection function from $\mathcal{F}_b$.** In Fig. 6 (Right), we train on compositions of two functions, where one position is one of 25 bijections, and the other

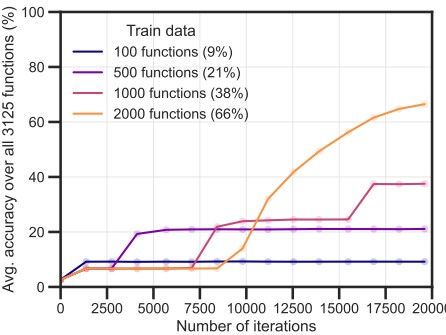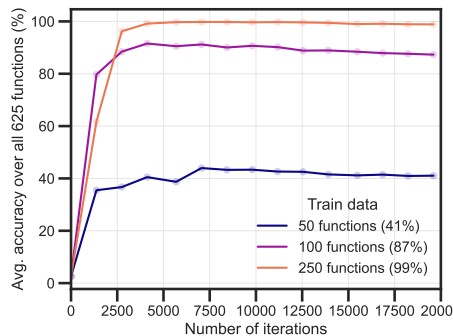

Figure 6: **Compositional generalization is less frequently seen in the direct prompt format compared to the step-by-step prompt format. (Left.)** We train a Transformer on 20+1 bijections with 5 compositions with 4 choices at each position. The model fails to generalize to all 3125 compositions even if it trained on 2000 such functions. **(Right.)** We train a Transformer on a composition of two functions, with one function being one of 25 bijections and the other function being one of 25 permutations (totalling to 625) compositions. The model is able to compose previously unseen combinations of functions when trained on 250 of these functions in this scenario.

is one of 25 permutations. We vary the number of compositions seen in the training data and find that 250 compositions in the training data is enough for the model to generalize to all 625 possible compositions of the two functions. We note that bijection and permutations operate on orthogonal features of the input: bijections operate on the value of the token while permutations operate on the position of the token. We speculate that this is important for compositional generalization in the direct format. Direct formatted prompts occur less frequently compared to step-by-step compositions and this could be indicative of why chain-of-thought is a popular prompting strategy (Wei et al., 2022b). A precise answer for when direct prompts can succeed remains unclear though.

**Why is out-of-order generalization harder for direct prompting?** We believe that direct prompts are unlikely to generalize to the out-of-order compositions or at least require more samples. For example, consider functions $F$ and $G$ and consider a Transformer that computes the function $G \circ F$. Since $G \circ F$ is computed using a single forward pass through a Transformer for direct prompts, $G$ must occur in a layer after $F$ (shown in Fig. 11b). As a result, the model cannot generalize to $F \circ G$ since $f$ occurs after $G$ in its layers. Hence, a Transformer may have to learn copies of $F$ and $G$ at multiple layers in order to generalize to both $F \circ G$ and $G \circ F$.

## 4.4 ANALYZING TRAINED TRANSFORMERS

**Linear probe accuracy.** In Fig. 7 (Left), we use a linear probe to analyze the importance of attention layers and contrast them with the MLP layers. Follwing Geva et al. (2022), we fix the parameters of probe to the last linear layer, i.e., the unembedding layer of the trained model. We use a Transformer trained on 100 **random** in-order compositions of 5 functions identical to the model in Fig. 4a. See Figure 14 for more linear probe experiments on Transformers of different sizes.

**Attention Visualization.** In Fig. 7 (Right), we plot the attention map for a predefined composition of functions from the set $\mathcal{F}_b$. Specifically, we take a pretrained 1-layer Transformer, which, as we show in Appendix, is able to solve at least the in-order generalization task. Then, keeping the Task tokens to be fixed corresponding to the predefined composition, we sample 1000 data tokens and compute the attention map for the 1-layer model. The average of these maps is reported in the figure: we clearly see that all data tokens attend to the Task token that specifies the current function that needs to be applied and the data token that the function is to be applied to. This provides further credence to our claim that attention is playing a non-trivial role in enabling compositionality.

**Training dynamics.** In Fig. 8, we consider a fine-grained version of Fig. 4a to understand if a Transformer can generalize to composition of fewer functions before it generalizes compositions of many functions. We find that the answer depends on the nature of the training data. If the training data consists of **21 base** and very few in-order compositions, then a Transformer generalizes to fewer compositions (more identities) first before generalizing to compositions of multiple functions. On the other hand, if the model is trained on 25 **random** in-order compositions, then it is better at generalizing to more complex compositions of these functions; this trend is lost when we train on 50 **random** in-order compositions.

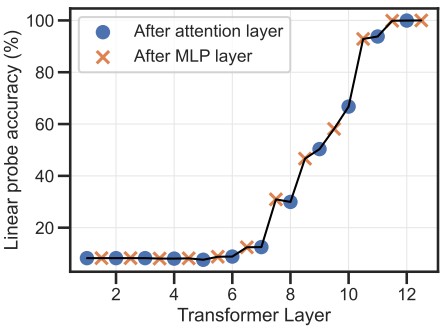 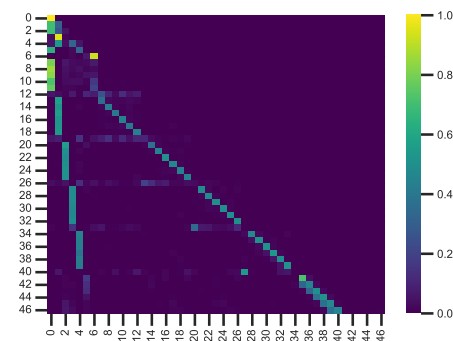

Figure 7: **(Left.) We see a sharp increases in accuracy after MLP layers in the last few layers of the Transformer.** We compute the linear probe accuracy—averaged over in-order compositions of functions—after the MLP and attention layers at every layer of the model. **(Right.) Attention is primarily paid to the relevant data and task token.** We plot the causal attention mask of a 1-layer Transformer trained using the step-by-step format on compositions of 5 in-order bijections (setup of Fig. 4). Keeping the prompt fixed to specify a specific composition of functions, we average the attention maps for 1000 samples. We clearly see a given data token attends to the specific task and data token relevant to producing the right output.

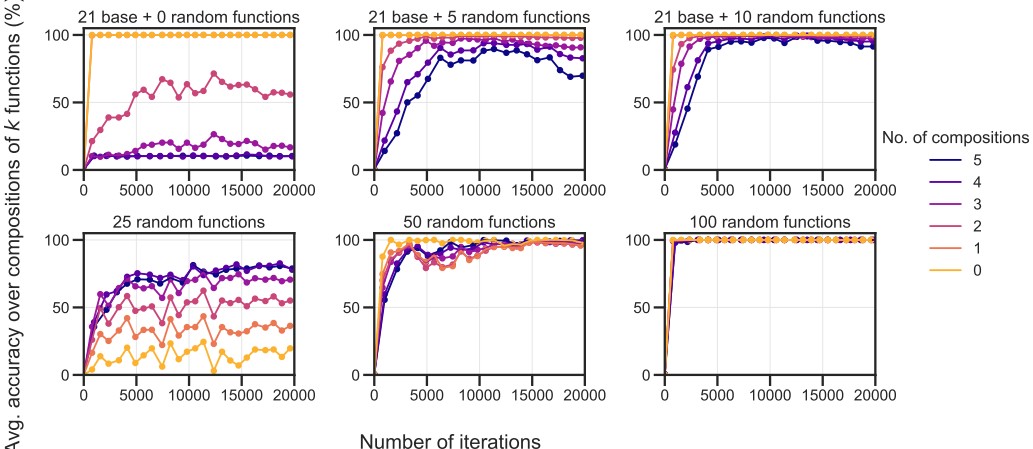

Figure 8: **A Transformer trained on a random subset of functions generalizes first to a composition of more functions before it generalizes to a composition of few of them.** Each line is the average accuracy over all composition of $k$ functions and each subplot is a Transformer trained on a different subset of functions. The **21 base** is trained on the individual functions and these Transformers learn to compose a smaller set of functions (more functions in composition are identity) before learning to compose many of them. The opposite is true when the model is trained on a random subset of 25 compositions of functions.

## 5 DISCUSSION

Given several recent works focused on prediction or elicitation of capabilities in pretrained models, we ask whether the very motivation guiding these works is tractable: can we possibly characterize all capabilities of a model, specifically a Transformer, pretrained on a compositional data domain? To address this question, we proposed a synthetic, but well-defined, data domain and formalized the notion of a capability as representing a function defined over the domain. Breaking compositional generalization into two relevant scenarios (in-order vs. out-of-order), we showed that the compositional structure of the data forces a model to learn to compose at relatively minimal data diversity, which indicatively address our primary question: an appropriate prompt could make the model compose its capabilities, yielding an "explosion of capabilities". This can arguably make tractable analysis of capabilities in a pretrained model relatively difficult.

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

# A    EXPERIMENTAL DETAILS

## A.1    TRAINING METHODOLOGY

**Transformer architecture**    We train variants of nanoGPT[1] with 12 layers, 12 attention heads and an embedding dimension of size 120. Each transformer block contains a causal attention layer, layer-norms, residual connections and an MLP (see Fig. 9).    The MLP contains two fully-connected layers sandwiched by a GELU layer (Hendrycks & Gimpel, 2016) The first fully-connected layers has a hidden layer with size 4 times the embedding dimension (480) and the second hidden layer has a size equal to the embedding dimension (120).

The input tokens are converted to one-hot vectors before being passed through to the Transformer.    The model makes use of no dropout and no biases in the Layer norm layers. We use weight-tying (Press & Wolf, 2016) in the Transformer which uses shared weights for the input and the output embedding layers. Finally, we make use of mixed-precision (bf16 in torch) to speedup training.

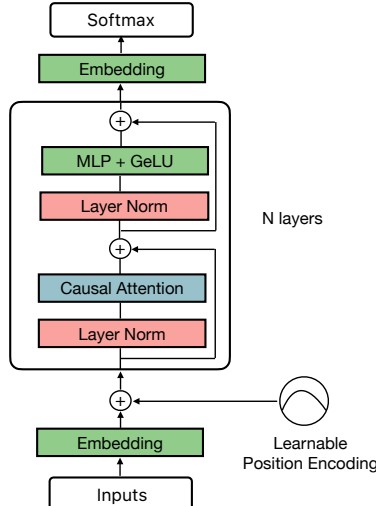

Figure 9: We use nanoGPT as the Transformer architecture in all our experiments. The core Transformer block is a layer norm, a causal attention block, followed by another layer norm and a 2-layer multi-layer perceptron (MLP). The Transformer block has two residual connections.

**Loss and Optimizer**    Models are trained using an autoregressive objective to predict the next token using the cross-entropy loss. Give a sequence of tokens of $t$ tokens denoted by $x_{1:t}$. Let $p_w(y \mid x_{1:t})$ denote the probability distribution over the next token as predicted by a model with weights $w$. For a sequence $x_{1:T}$ of length $T$, the autoregressive objective is

$$L(w) = \sum_{t=1}^{T-1} log p_w \left( y = x_{t+1} \mid x_{1:t} \right).$$

Trained is performed for 100 epochs with a cosine-annealed scheduled with warmup. We use an initial learning rate of 3e-4 annealed eventually to 6e-5. We use AdamW as the optimizer ($\beta_1 = 0.9$ and $\beta_2 = 0.95$) with a weight decay 1e-3 and a batch-size of 512. We also make use of gradient clipping with a magnitude of 1.

## A.2    DATA GENERATING PROCESS

**Data and task tokens.**    Both data and task tokens are converted to one-hot vectors before being fed to the Transformer. The set of data tokens is denoted by $X_d$ and the vocabulary $|X_d|$ is of size 10 in all our experiments. The data tokens in the input $x_d \in X_d^6$ is a sequence of 6 tokens and is the input to the function composition. The 6 tokens are sampled uniformly at random from $X_d$ with replacement.

There are two sets of functions considered in this work. The set of functions $\mathcal{F}_b$ (which we refer to as bijections) applies a lookup table in an element-wise fashion to each of the 6 tokens in $x_d$. The set of functions in $\mathcal{F}_p$ permute the 6 tokens in $x_d$. The family of functions in $\mathcal{F}_b$ and $\mathcal{F}_p$ are described in Fig. 10. Each function from $\mathcal{F}_p$ and $\mathcal{X}_b$ has its own task token in $X_F$.

The input starts with a sequence of $L$ task tokens $x_f \in X_F^L$. The number of compositions is $L = 2$ in experiments like Figs. 15, 6 (Right) while most other experiments use $L = 5$ compositions.

---

[1]https://github.com/karpathy/nanoGPT

**Sampling task tokens** The task tokens can be sampled such that they satisfy certain properties. For example, let us consider the composition of two functions – one from the set $\mathcal{F}_1 \subset \mathcal{F}_p$ and another from $\mathcal{F}_2 \subset \mathcal{F}_b$ (which is the setting in Fig. 6 (Right)). We can restrict the training data to compositions from the set $\mathcal{F}_2 \circ \mathcal{F}_1$ which are in-order compositions (see Fig. 2). Alternately, we can also choose to include out-of-order composition which include compositions from $\mathcal{F}_1 \circ \mathcal{F}_1, \mathcal{F}_2 \circ \mathcal{F}_2$ and $\mathcal{F}_1 \circ \mathcal{F}_2$. In Fig. 6 (Right), we restrict our training and evaluation to in-order compositions of functions and we observe that training on a subset of the elements from $\mathcal{F}_2 \circ \mathcal{F}_1$ suffices to compositionally generalize all functions in the set.

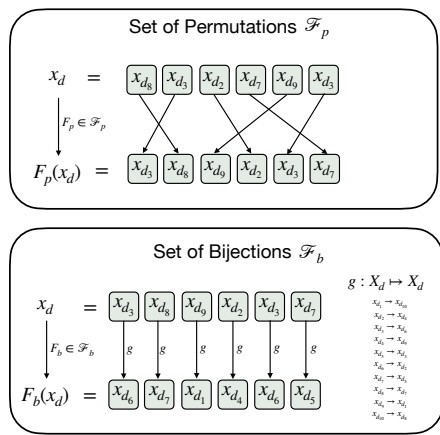

Figure 10: A permutation from $\mathcal{F}_p$ permutes the 6 tokens in the input $x_d$. A bijection from $\mathcal{F}_b$ applies a lookup table to each of the 6 tokens individually.

Two other commonly used subsets of functions are **21 base** and **random**. Consider $\mathcal{F}_1, \mathcal{F}_2, \ldots, \mathcal{F}_5 \subset \mathcal{F}_b$. The set **random** considers $k$ functions from the set $\mathcal{F}_5 \circ \mathcal{F}_4 \circ \cdots \circ \mathcal{F}_1$ which are drawn uniformly at random.

**21 base** is used to test if the compositionality is seen when the Transformer is trained on the individual functions from $\mathcal{F}_i$ for all $i \in [5]$. In the training data, all compositions have 4 of the 5 functions to be the identity function $I$, i.e it considers compositions of the form $I \circ I \circ \mathcal{F}_3 \circ I \circ I$ or $I \circ \mathcal{F}_4 \circ \cdots \circ I$. There are a total of $1 + \sum_{i=1}^{5} \mathcal{F}_i$ such functions; the 1 is when all 5 functions in the composition are identity. The model is never trained on the composition of two or more functions, and at least compositions of 3 functions are necessary to generalize to all in-order compositions Fig. 19.

**Generating a sequence of tokens** A sequence starts with a sequence of two task tokens $x_f = [x_{F_1}, x_{F_2}]$ followed by a sequence of data tokens $x_d$. The sequence can either be presented in the step-by-step format (Figure 11a) where the intermediate outputs are also included in the sequence. For example, the sequence in the step-by-step format would look like $[x_{F_1}, x_{F_2}, x_d, F_1(x_d), F_2(F_1(x_d))]$. The direct format (Figure 11b) does not include the intermediate outputs of the composition in the sequence and an example of such a sequence is $[x_{F_1}, x_{F_2}, x_d, F_2(F_1(x_g))]$.

The step-by-step and direct formats are also discussed in Fig. 3. The training data consists of 100,000 sequences for all experiments in one of the two formats.

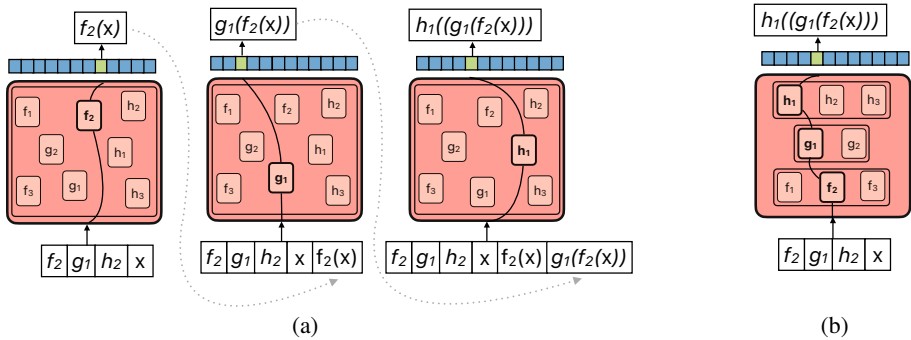

(a)                                                                                        (b)

Figure 11: **Step-by-step composition v.s. Direct composition.** We test two possible routes for compositions. **(a)** Step-by-step prompting, which allows for generating intermediate outputs. **(b)** Direct prompting, where the model must compose the functions without the intermediate outputs.

**Evaluating compositions** When evaluating trained models, we evaluate on 1000 different inputs for every composition of functions. Since Fig. 5 requires us to evaluate on a combinatorial set of

functions, we sampled 1000 functions (or the total number of functions, whichever was lower) for each cell which can be identified by the displacement and number of compositions, and we compute the accuracy averaged over those functions to populate the cell. The accuracy of a completion is calculated by averaging the accuracy of the last six tokens. We see that qualitative trends do not change when we use different metrics Figure 20.

**Computing linear probe accuracy** We consider the outputs after every attention block and every MLP block (including the residual stream in both cases). We then pass these outputs through the final embedding layer and a softmax layer to get predictions over the next token. We use these predictions to compute the accuracy at that layer. The accuracy is averaged over 1000 different input data tokens and for 200 different compositions of functions.

## B ADDITIONAL EXPERIMENTS

### B.1 SWEEPING HYPER-PARAMETERS OF THE TRANSFORMER

We vary the number of layers, the number of attention heads. and the embedding dimension of the nanoGPT model in Fig. 13. We consider the setup identical to Fig. 4; all models are trained on 50 **random** in-order compositions of 5 bijections. We report accuracy averaged over all 3125 in-order compositions.

We make the folllwing observations: (1) Most surprisingly, the accuracy reduces as the number of layers become *huge* for this compositional task; we expect that this is due to issues with optimization of a large depth model. (2) The accuracy does not change with the number of attention heads for a 1-layer Transformer. (3) The accuracy increases as we increase the embedding dimension and the model under fits the training data when the embedding dimension is too small.

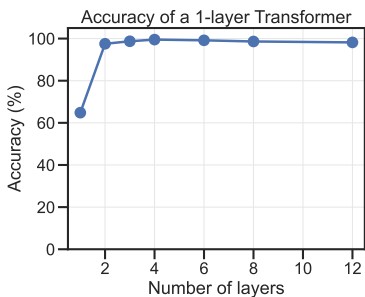

Figure 12: **Transformers requires at least 2-3 layers for compositional generalization with the direct prompt format.** We vary the number of layers in the Transformer and train on direct composition in a setup identical to Fig. 6 (Right).

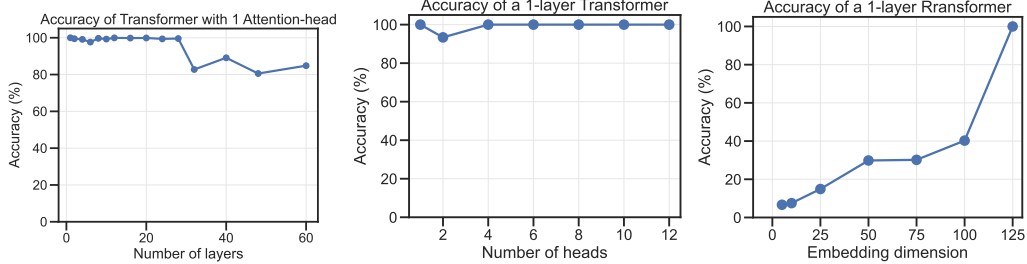

Figure 13: **We see compositionality in Transformers even if we change the number of layers and attention heads.** Compositionality is seen even in a 1-layer Transformer when trained with the step-by-step prompt format on 50 in-order compositions of bijections. However the ability to compose degrades as we increase the number of layers in the Transformer.

### B.2 LSTMS DO NOT LEARN TO COMPOSE

We report results on autoregressively trained LSTMs using the step-by-step prompt format in Table 2 and using the direct prompt format from Table 1. **LSTMs fail to generalize outside of the training data while Transformers generalize compositionally in both these scenarios**. This points to an inductive bias that helps Transformers trained with an autoregressive objective generalize.

The LSTMs are trained using the same data using the autoregressive objective defined in Appendix A. We use the AdamW optimizer with learning rate equal to 3e-4 ($\beta_1 = 0.9$ and $\beta_2 = 0.95$), batch size of 512 and weight decay of 1e-4 for 150 epochs. As is common, we do not use a positional embedding, since the architecture is not permutation invariant. The inputs are passed through an input embedding layer before being passed to the LSTM and the outputs of the LSTM are also passed through a linear layer which outputs the logits. In our experiments, we vary the number of stacked LSTMs (or no. of layers) and the dimension of the internal hidden vector.

Despite our attempt to train multiple different LSTMs with the best set of hyper-parameters, we observe that they do not show any compositional generalization on all our synthetic setups. This observation is further evidence for our hypothesis that the attention layers are important for compositionality.

|        | Hidden dimension | |
| Layers | 256 | 5124 |
|---|---|---|
| 1 | 22.5 | 46.0 |
| 2 | 33.4 | 69.1 |

Table 1: **LSTMs fail to compose in the direct prompt format.** We train an LSTM on 250 composition of two functions (one permutation and one bijection) in the direct prompt format and tabulate the accuracy (%); the setup is identical to Fig. 6 (Right).

|        | Hidden layer dimension | | | |
| Layers | 120 | 256 | 512 | 1024 |
|---|---|---|---|---|
| 1 | 16.2 | 36.2 | 99.9 | 99.9 |
| 2 | 60.3 | 99.3 | 99.9 | 99.8 |
| 4 | 18.7 | 100.0 | 100.0 | 9.9 |

|        | Hidden layer dimension | | | |
| Layers | 120 | 256 | 512 | 1024 |
|---|---|---|---|---|
| 1 | 9.3 | 10.3 | 20.1 | 22.9 |
| 2 | 12.4 | 21.3 | 25.3 | 28.8 |
| 4 | 6.6 | 13.9 | 17.6 | 10.0 |

Table 2: **LSTMs fail to compose in the step-by-step prompt format.** We train autoregressive LSTMs on 50 in-order compositions of 5 bijections from $\mathcal{F}_b$ in the step-by-step format and tabulate the accuracy (%); The setup is identical to Fig. 4. We evaluate the LSTM on the **(left)** compositions seen during training and **(right)** in-order compositions not seen during training. LSTMs fail to generalize to functions outside of the training data while transformers generalize compositionally in the same setting.

## B.3 ATTENTION MASKS

**Detailed setup.** We train a 1-layer Transformer on a composition of 50 **random** in-order compositions of 5 bijections in the step-by-step prompt format. We visualize the attention masks for a fixed sequence of task tokens, averaged over 1000 different data tokens in Fig. 7(right). We found the attention masks to be identical across different choices of the task tokens. Each row corresponds to a causal attention mask for a single token and sums up to 1. At any given row, the attention is over two elements which we speculate are the task token and the intermediate output of the composition. The 5 contiguous blocks along the columns correspond to the 5 steps of composition. These preliminary results indicates that it is possible to build a complete mechanistic understanding of Attention for compositional tasks.

## B.4 PROBING THE LAYERS IN TRANSFORMERS OF DIFFERENT SIZES

In this section, we consider an experimental setup that is identical to the linear probe experiments in Figure 7. We compute the probe accuracies for Transformers with different number of layers in Fig. 14. Across all Transformers, we observe that accuracy increases in the last few layers of the transformers. Furthermore, we also observe a sharp increase in accuracy right after the MLPs in the last few layers of the transformer.

We saw in Figure 7(right) that the attention masks for a 1-layer Transformer seem to select an input and a task token to operate on at every step of the composition. We hence believe that attention has a huge role in compositionality and propose the following hypotheses: (1) LSTMs fail to compose functions not present in the training data. We hypothesize that a lack of attention contributes to this failure. (2) The probe accuracy after some MLPs see a sharp in increase in accuracy. We hypothesize that the attention layers play a critical role in selecting the right inputs to pass to the MLP.

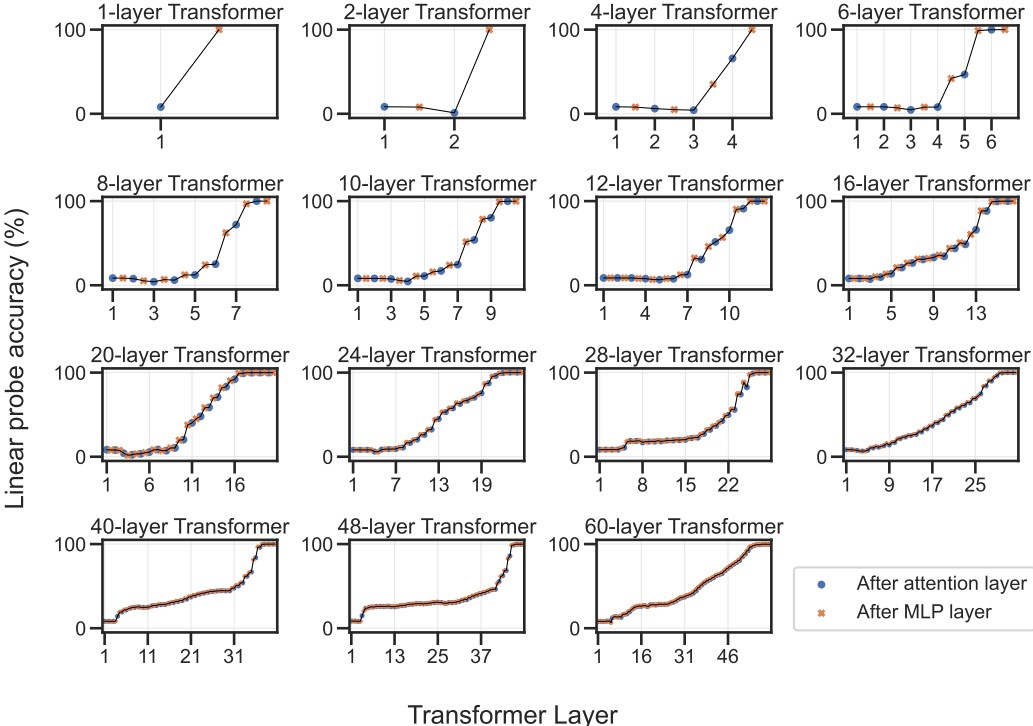

Figure 14: **We use a linear probe to study the accuracy at different layers on Transformers of different sizes.** Most architectures see an increasing in accuracy in the latter half of the Transformer. The increase in accuracy is more gradual for Transformers with more layers. The accuracy increases sharply after an attention layer across all architectures.

## B.5 ANOTHER FAILURE WITH THE DIRECT FORMAT WITH BIJECTIONS

In Fig. 6 (Left) we show that Transformers do not learn to compose 5 bijections and only generalize to compositions in the training data. Figure 15 augments this result and shows that a similar failure occurs even when we consider the composition of just two bijections. Hence the model may not compose some function in the direct prompt format and the step-by-step format with an autoregressive objective is far more amenable to compositions.

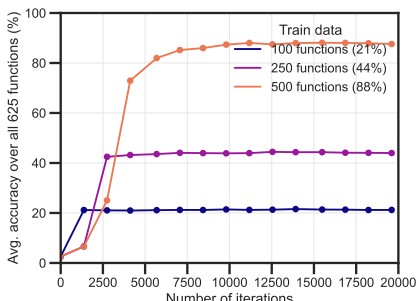

Figure 15: **Transformers fail to generalize to compositions of even 2 bijections, when trained with the direct prompt format.** The curve depicts the accuracy over all 625 in-order compositions of two bijections (25 choices for each bijection) when trained on different subsets of in-order compositions. The model is trained with direct composition. Even if we train on 500 such compositions, the model fails to generalize to the remaining 125 compositions. This is additional evidence that the model is incapable composing bijections through direct composition.

## B.6 Additional experiments with training data from RANDOM and 21 BASE

In this section, we conduct a collection of analyses for a model trained on in-order compositions of 5 bijections in the step-by-step prompt format. We (1) Compare how **21 base** and **random** generalize to other in-order compositions; (2) Test if the compositions are systematic; (3) Look at alternate evaluation metrics (4) Change the number of **random** functions in the training data; (5) Limit the maximum number of compositions in the training data and evaluate compositional generalization.

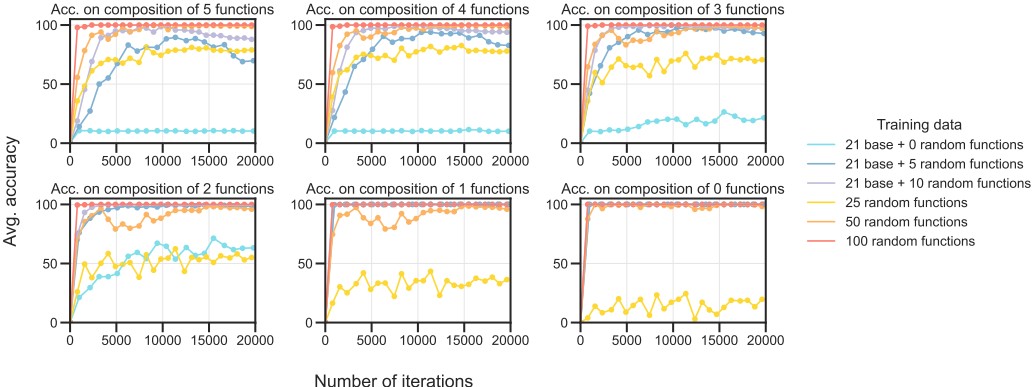

Figure 16: **How do different training datasets generalize to compositions of many and few functions?** This is a fine-grained version of Fig. 4a. Model trained on 50 **random** compositions generalizes poorly compositions of small number of functions while a model trained on the **21 base** generalizes poorly to composition of 4 or 5 functions.

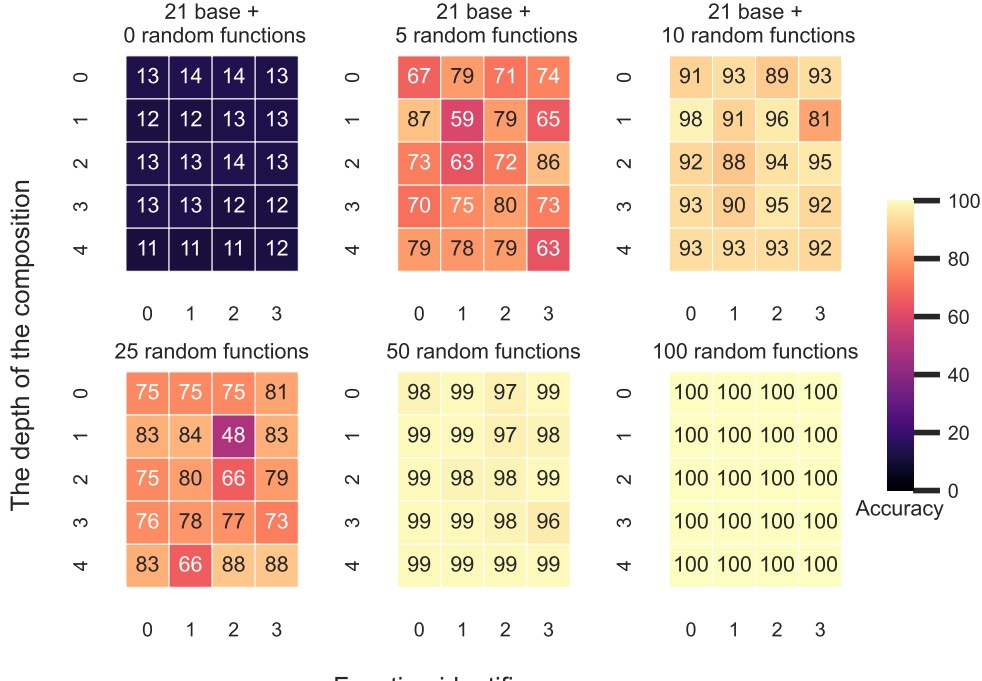

Figure 17: **Systematicity.** We consider trained models from Fig. 4a and analyze the accuracy of each of the 20 functions (monolithic capabilities) when averaged all instances in which it was used compositionally. We breakdown the results to see if certain functions are more accurate when used in compositions compared to others and find that models seem to learn all functions equally well.

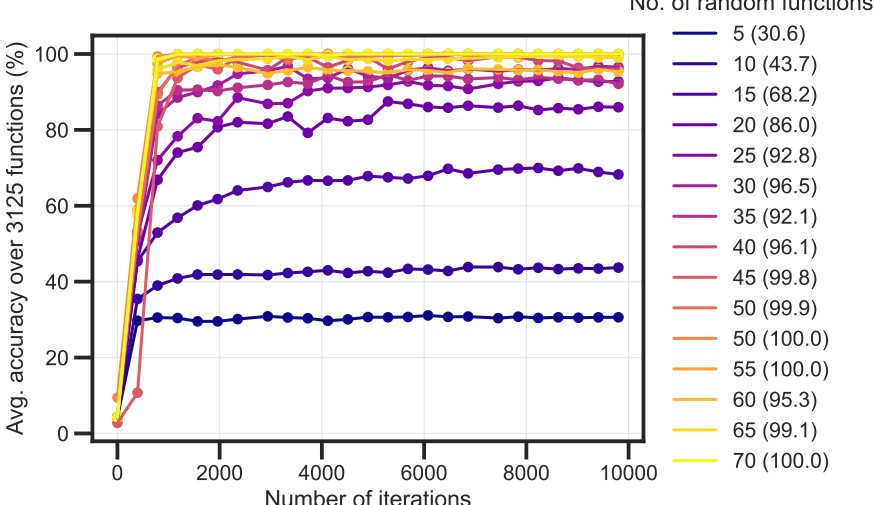

Figure 18: **Training with different numbers of random functions.** We train on a different number of random functions ranging from 5-70 in steps of 5. These plots are the accuracies averaged over all in-order compositions of 5 bijections over the course of training.

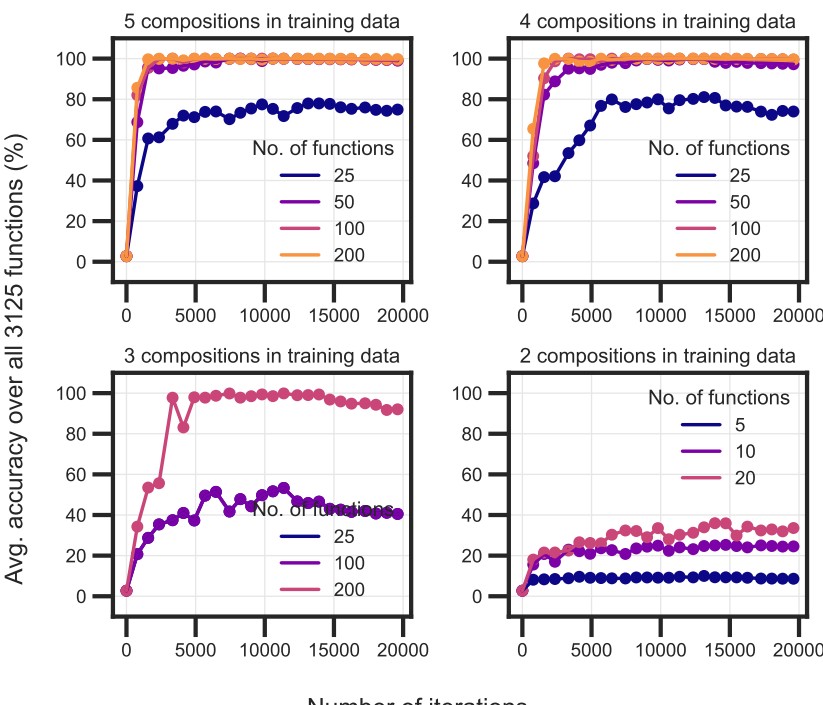

Figure 19: **Limiting maximum number of compositions in the training data.** The figure plots the accuracy on all in-order compositions against the number of training iterations. Each sub-plot considers compositions of size exactly 2, 3, 4, 5, respectively in the training data. The model is able to generalize to most in-order compositions only if the training data consists of compositions of size at least 3 (bottom-right).

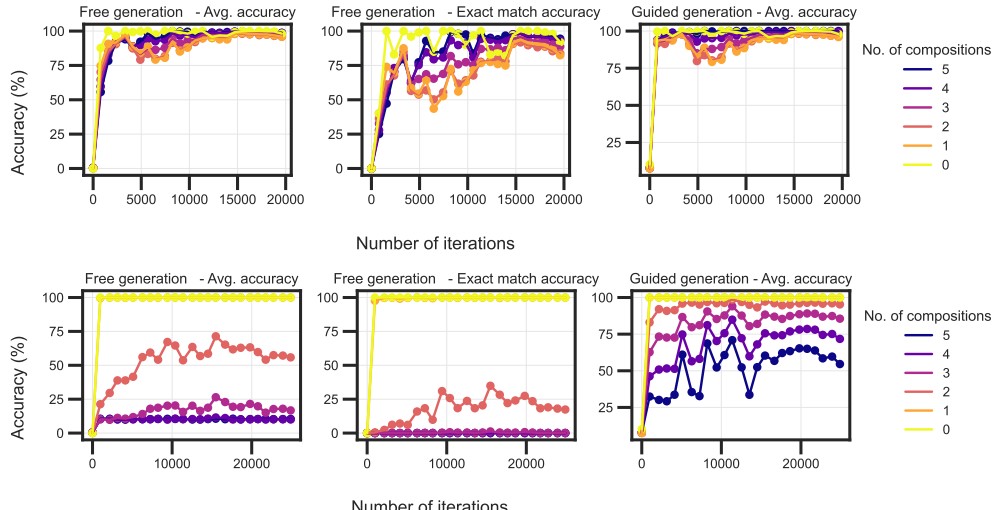

Figure 20: **Evaluation metric.** We consider 3 different metrics for evaluating the models. The left column considers the average accuracy when the model generates **The choice of metric doesn't change qualitative trends.** Each sub-plot considers compositions of only size 2, 3, 4, 5, respectively. In each plot, we vary the number of such functions that are present int he training data. **One exception is when we train on compositions of size 2.** In this case, the guided generation accuracy is high, but the free generation accuracy is not.

## B.7 WORD EMBEDDINGS

We study the token embeddings of the Transformer models and observe that they are similar for models with different number of layers and attention heads. We notice a block diagonal structure that separates task tokens from the data tokens. We also observe another block diagonal structure within the task tokens which occurs when we train only on in-order compositions.

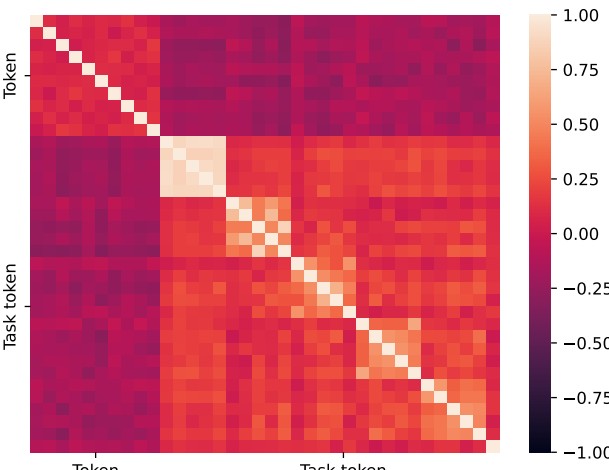

Figure 21: **Word embedding correlations present a block-diagonal structure that separates data tokens from task tokens.** We plot the inner product between all pairs of word embeddings of the tokens. The task tokens are orthogonal to the set of input tokens. Different functions in the same level, i.e. $\{F_i^{(l)}\}_{i=1}^N$ for a fixed $l$, form a block-diagonal in this matrix. We observe similar word embeddings in Transformers of different sizes.

