# OpenReview forum: "How Capable Can a Transformer Become? A Study on Synthetic, Interpretable Tasks"
_ICLR.cc/2024/Conference — Submitted to ICLR 2024_

### Official Review · Reviewer_xno8 · 2023-10-24

**Soundness:** 3 good
**Presentation:** 2 fair
**Contribution:** 4 excellent
**Rating:** 8
**Confidence:** 4

**Summary:**

The paper investigates the capabilities of transformers to learn individual functions and compose them in a sequential manner. A major contribution of the study is the introduction of a synthetic dataset that is both interpretable and straightforward to implement.
While the paper is mostly well-written, it falls short in clearly describing the experimental setup.



Overall, the paper is commendable for its innovative synthetic dataset, which provides valuable insights into the capabilities of transformers for function composition. However, the paper would benefit from greater clarity in its experimental descriptions and more robust support for some of its conclusions.

**Strengths:**

- The dataset introduced is both simple and valuable for future research.
- The paper presents perceptive experimental analyses, notably the discussion on the differences between in-order and out-of-order generalizations.

**Weaknesses:**

- The experimental setup lacks sufficient clarity, making it challenging to understand the specific procedures.
- The paper's conclusion regarding the significance of later attention layers is not adequately substantiated by the experimental data; it lacks insight.

**Questions:**

- Section 3.2 lacks a clear definition and differentiation between permutations and bijections. It should be explicitly stated that the results in Section 4.1 pertain only to bijections.
- Figure 3 needs more explanation about the notations. For instance, what $f_{i-j}$ means?
- The explanation between "random" and "21 base" in Section 4.1 is unclear.
- The bottom-left subplot in Figure 7 exhibits an anomalous trend compared to other subplots; could you please explain this? In particular, there is no clean trend that transformer first learn composition of more functions.
- The experimental setup for Figure 6(b) in Section 4.3 is not elucidated. What is the rationale for having 25 bijections and 25 permutations?
 - The paper claims that adding permutations aids direct composition; this is a counterintuitive finding that warrants further elaboration.
- Figure 6(b) requires more detailed explanation, and several typos need rectification.


Typos:
 - In page 2, remove one of the "considered" and the quotation masks are not correct.
 - in Figure 11: "direction composition" should be corrected to "direct composition."

---

> ### Author Response · Authors · 2023-11-21
> **Rebuttal to Reviewer xno8 (1/2)**
>
> We thank the reviewer for their positive feedback. We are glad that the reviewer finds our synthetic setup to be interpretable and useful for future research, the insights on Transformer capabilities to be valuable, and our experimental analyses to be perceptive. We have addressed the questions and suggestions below. We hope the reviewer champions the acceptance of this paper.
>
> **>>> The paper's conclusion regarding the significance of later attention layers is not adequately substantiated by the experimental data; it lacks insight.**
>
> We have added two new experiments to further substantiate this claim.
>
> In the first experiment, we visualize the attention maps of a Transformer on the step-by-step prompt. In Figure 7(right), we observe that the attention map selects two tokens, 1 task token (which picks the function) and 1 input token which is the intermediate result of a composition. Hence, attention seems to play an important role in selecting the function and the input that the function should be applied to.
>
> In the second experiment, presented in Appendix B.4 (figure 14), we study the attention layers of transformers of different depths (1 to 60 layers). Across most Transformers, we find that compositions occur in the later layers of a Transformer. We also observe that a sharp upward inflection in accuracy in all the plots occurs after an MLP layer. The attention layers seem to select the input and the function to apply and the computation seems to occur in the MLP.
>
> Despite the strong evidence, these claims are still speculative and we have made sure to convey the same in the manuscript.
>
> **>>> The bottom-left subplot in Figure 7 exhibits an anomalous trend compared to other subplots; could you please explain this? In particular, there is no clean trend that transformer first learn composition of more functions.**
>
> Each line corresponds to the accuracy averaged over composition of a fixed number of functions. The purple line corresponds to the average accuracy, where we evaluate all compositions of 5 functions, while the orange line for compositions of 1 function (the other 4 are identity). Note that compared to the top-middle plot, the colors are reversed in order. The bottom-left plot highlights that at the end of training on 25 random functions, the model has a harder time generalizing to compositions of fewer functions compared to compositions of many of them (yellow and orange lines are always below purple).
>
> **>>> The paper claims that adding permutations aids direct composition; this is a counterintuitive finding that warrants further elaboration.**
>
> We believe there is a misunderstanding and would like to clarify that this is not the exact claim that we have made in the paper. We restate our result below.
>
> The direct prompt format is successful if we train on compositions of two functions: a permutation function and a bijection function. Hence the function composition is an element from the set $\mathcal{F}_b \circ \mathcal{F}_p$. It is surprising that direct prompt format fails when we compose 2 functions which are both bijections, i.e. Transformers fail to generalize to compositions in the set $\mathcal{F}_b \circ \mathcal{F}_b$.
>
> We do not have a precise answer for why this is the case and find this observation surprising. But we speculate why this could occur. Permutations operate on the position of a token and bijections operate on the value of a token; the position and value are (nearly) orthogonal features of the input. Okawa et al., Lewis et al. (https://arxiv.org/abs/2310.09336, https://arxiv.org/abs/2212.10537) make similar observations for vision models and show compositional generalization on an orthogonal set of attributes. We believe a similar phenomenon is at play when a Transformer is forced to compose in a single step.

---

> ### Author Response · Authors · 2023-11-21
> **Rebuttal to Reviewer xno8 (2/2)**
>
> ## Improvements to writing
>
> We thank the reviewer for the careful reading of our work. We have incorporated the changes to the manuscript and have highlighted them in green.
>
> **>>> The experimental setup lacks sufficient clarity, making it challenging to understand the specific procedures.**
>
> Thank you for raising this concern. We have significantly expanded Section 3.1 and Appendix A to include more details. We also commit to releasing the code.
>
> **>>> Section 3.2 lacks a clear definition and differentiation between permutations and bijections. It should be explicitly stated that the results in Section 4.1 pertain only to bijections.**
>
>
> We have added details to the section (we note it is now section 3.1) and to appendix A.1 where we describe the set of permutations and bijections. We have also clarified in sub-section 4.1 that the results only pertain to bijections.
>
> **>>> Figure 3 needs more explanation about the notations. For instance, what f_{i_j} means?**
>
> We have tried to explain the notation further in the updated manuscript and have also simplified it for ease of understanding. Specifically, Figure 2(a) now explains the role of $i$ and $j$ for a function $F^{(i)}_j$. The superscript $i$ determines the position in the composition for an in-order composition. For example in a composition of 3 functions, the function $F^{(2)}_3$, must be in the second position for an in-order composition, i.e., an in-order composition of 3 functions should look like $g \circ F^{(2)}_3 \circ h$ since $i=2$.
>
> The index $j$ is used to iterate over the set of all possible functions that are allowed to be at position $i$. For example $F^{(2)}_1$, $F^{(2)}_2$ and $F^{(2)}_3$ (j=1,2,3) are all functions that can appear in the second position in an in-order compositions. All of these functions have the same color in Figure 3.
>
> We have improved the writing in Section 3.1 to complement Figure 3. We would be glad to further clarify if the reviewer believes that the details are still not entirely clear.
>
> **>>> The explanation between "random" and "21 base" in Section 4.1 is unclear.**
>
> Thank you for letting us know. We have added a section to Appendix A.1 explaining the 2 subsets in more detail.
>
> The in-order compositions of 5 functions are depicted in Figure 2(a). This figure will be useful for understanding the set of functions “random” and “21 base” which are both subsets of the set of all bijections $\mathcal{F}_b$.
>
> We consider the composition of 5 functions where each position can take one of 5 possible functions with one of the choices being identity. The two sets are defined as follows:
>
>
>
> 1. Base 21:  Set of all functions such that at least 4 of the 5 positions are identity functions. There are a total of 21 such compositions in this set and they can be used to generate every other composition.
> 2. Random: Set of 25 compositions sampled uniformly at random from the set of all in-order compositions (3125 such compositions)
>
> “21 base” does not have compositions of two or more functions unlike “random” and forms a basis for the group of all in-order compositions.
>
> **>>> The experimental setup for Figure 6(b) in Section 4.3 is not elucidated. What is the rationale for having 25 bijections and 25 permutations?**
>
> We would be grateful if the reviewer could clarify which specific details are unclear in Figure 6(b). The experimental setup (architecture, training methodology) is identical to all other experiments in the paper, as discussed in the beginning of the section. Specifically, Figure 6(b) considers the composition of two functions — one of which is a bijection and the other is a permutation.
>
> Since we consider the composition of only two functions, we increase the number of function choices at each position to 25. This results in a total of 625 compositions of functions. If we instead considered only 5 bijections and 5 permutations, then the total number of compositions is only 25, which would make our evidence for compositional generalization weaker.
>
> **>>> Typos in Page 2 and Figure 11**
>
> Thank you for bringing this to our attention.

---

### Official Review · Reviewer_1C3m · 2023-10-30

**Soundness:** 3 good
**Presentation:** 2 fair
**Contribution:** 1 poor
**Rating:** 3
**Confidence:** 4

**Summary:**

The paper presents an empirical study on how capable Transformer models are to generalize compositionally, a question that has been broadly discussed in the last 5 years in research literature. The specific task that the paper uses for the empirical study is composing functions defined on the domain of fixed-length works over a fixed vocabulary. The paper considers stepwise (i.e. with intermediate outputs) and direct composition (i.e. without intermediate outputs) setups. Another axis variation is whether the function order can be different at test time than training time (in-order vs out-of-order). The key observations are that (1) provided enough diversity in the training data, Transformers can learn to compose functions (2) stepwise composition is easier to learn.

**Strengths:**

The paper is mostly clearly written and easy to understand.

**Weaknesses:**

I don’t think that this paper makes a meaningful contribution to the field of deep learning on top of the already available work. The general question of whether Transformers or other neural architectures can generalize compositionally has already been discussed in numerous papers, including similar function compositional setups ([1, 2]). The general consensus in the literature is that with enough diversity, any neural architecture can learn any compositional behaviors. Further interesting questions can be asked: what architectures can learn compositional behavior from less diverse training data, can we get diverse enough training data to achieve compositionality in real-world tasks (GPT-4 seems to partially answer that), how compositional generalization abilities of neural models compare to those of humans. This paper, however, does not go deeper into one of these or any other direction, it discusses compositional generalization and the highest most abstract level, at which the answer is: it depends. While the paper acknowledges that there is ample prior work on the topics, the paper fails to explain what it adds on top. The finding that step-wise composition is easier than direct is rather unsurprising, especially in the view of chain-of-thought prompting of LLM that has been getting popular lately.

[1] https://arxiv.org/abs/1802.06467
[2] https://aclanthology.org/2022.emnlp-main.662.pdf

**Questions:**

- Section 3.2 is a bit unclear on what are “bijection” and “permutation” mappings in the paper’s context. My understanding is that bijection here means per-token bijection, whereas permutation means shuffling tokens of the word.
- The details of what the vocabulary size and what the size of the word is were difficult to find.

---

> ### Author Response · Authors · 2023-11-21
> **Response to Reviewer 1C3m (1/3)**
>
> We thank the Reviewer for their feedback. We hope to engage in a discussion and convince the Reviewer of the importance of our work. We have answered specific questions below.
>
> **>>> Further interesting questions can be asked ………..**
>
> All these questions are indeed very interesting. However, we emphasize that our work has a very different goal. We do not intend to develop novel methods for improving Transformers’ ability to compositionally generalize. Our goals are twofold: 1) to demonstrate that compositionality can occur in autoregressively trained Transformers without changes to standard training pipelines, and if the underlying data domain itself is compositional, and 2) to understand what drives the existence of compositionality in large models. Importantly, we emphasize the motivation for our goals, as discussed in detail in the introduction, is grounded in recent literature on eliciting or predicting capabilities in pretrained models [1, 2]. Specifically, we aim to argue that if via training on a compositional domain (e.g., language), a model learns to compose its capabilities, then empirical benchmarking will be insufficient to characterize what capabilities the model possesses, i.e., what tasks it can perform.
>
> 1. Percy Liang, et al. Holistic evaluation of language models. arXiv preprint arXiv:2211.09110, 2022
> 2. Jordan Hoffmann, et al. Training compute-optimal large language models. arXiv preprint arXiv:2203.15556, 202
>
> **>>> How compositional generalization abilities of neural models compare to those of humans. This paper, however, does not go deeper into one of these or any other direction**
>
> We believe it is unfair that the reviewer asks us to tackle questions that compare Transformers to humans. We believe the results in our work are new and compelling and the suggestions seem entirely based on the reviewers interests, which we feel is very different to the motivations and goals of this work, as specified in detail in the introduction.
>
> We restate the contributions of our work in a broader context. There are many prior works that attempt to estimate the capabilities of Language models [1,2,3]. One aspect of understanding LLMs is to study inductive biases and properties of Transformer architectures on minimal synthetic setups, similar in spirit to [4, 5, 6]. Motivated by this, our work demonstrates that autoregressive Transformers (with commonly used training pipelines) learn to compose functions across several different scenarios; LSTMs fail on this same task. Our results on the synthetic task add credence to the hypothesis that language models can have far more capabilities than the ones directly present in the training data. Estimating these capabilities can be hard since the models need to be prompted appropriately—which is possible in our synthetic setup. To better understand why this occurs, we also study many variations to the properties of the training data and offer a preliminary mechanistic analysis.
>
> To address reviewer’s comment, we have explained this narrative more clearly in the revised version of the manuscript.
>
> 1. Aarohi Srivastava, et al. Beyond the imitation game: Quantifying and extrapolating the capabilities of language models. arXiv preprint arXiv:2206.04615, 2022.
> 2. Percy Liang, et al. Holistic evaluation of language models. arXiv preprint arXiv:2211.09110, 2022
> 3. Alex Tamkin, et al.. Understanding the capabilities,limitations, and societal impact of large language models. arXiv preprint arXiv:2102.02503, 2021.
> 4. Bingbin Liu et al., . Transformers learn shortcuts to automata. arXiv preprint arXiv:2210.10749, 2022.
> 5. Zeyuan Allen-Zhu and Yuanzhi Li. Physics of language models: Part 3.1, knowledge storage and extraction. arXiv preprint arXiv:2309.14316, 2023a.
> 6. Shivam Garg, et al.. What can transformers learnin-context? a case study of simple function classes. Advances in Neural Information Processing Systems, 35:30583–30598, 2022

---

> ### Author Response · Authors · 2023-11-21
> **Response to Reviewer 1C3m (2/3)**
>
> **>>> The general consensus in the literature is that with enough diversity, any neural architecture can learn any compositional behaviors.**
>
> We disagree with the reviewer’s claim that any neural architecture can learn compositional behaviors with enough diversity. We present a new result in the manuscript where we train an LSTM on the same task as used for our Transformer models in both the direct and step-by-step prompt formats. LSTMs perform significantly worse compared to Transformers (~30% vs ~100%), highlighting that **the architecture has a large impact on compositional generalization for the same training data**.
>
> **We would also like to point out that “enough diversity” is ill-defined and our work also attempts to characterize the same.** We study how precisely defined properties of the data affect downstream generalization: we show how factors like spurious correlations in the training data (in-order compositions), details of the prompt format (step-by-step and direct), and the number of compositions in the training data have vastly different effects on compositional generalization. Importantly, we emphasize that diversity in the data does not necessarily guarantee compositional behavior. In our work, we find that training on 2000 of the 3125 bijections (Figure 6 (left)) does not result in compositional behavior if we use the direct prompt format. However we can compositionally generalize using the direct prompt format if we use permutations and bijections. **Hence, our results indicate that there are different types of compositional data and Transformers can fail to learn some of them even with sufficient diversity.**
>
> Finally, we would like to add that **Transformers trained with the step-by-step data format are able to compose with very little diversity** in the training data. In Figure 5, we can train on as few as 125 functions (**0.003% of the total number of functions**) but we can still generalize to a combinatorial set of functions not present in the training data (4 million of them). This is a surprising result and points to how data diversity, while important, can be rather minimal when the appropriate architecture and data format are used.
>
> **>>> I don’t think that this paper makes a meaningful contribution to the field of deep learning on top of the already available work.**
>
> **>>> While the paper acknowledges that there is ample prior work on the topics, the paper fails to explain what it adds on top**
>
> **>>> including similar function compositional setups ([1, 2])**
>
> Thank you for pointing us to these papers. We have included them in our discussion in the Related work section. However, we hope to convince the reviewer of the importance of our work in the context of these prior works.
>
> Our work differs from prior work like [1, 2] (and several others that are reviewed in Section 2) in the following ways:
>
>
> 1. The goal of works like [1, 2] is to design new or improve existing architectures such that they compose better. While this is an important problem, we re-iterate that **our goal is different**: we would like to demonstrate that Transformers can learn to compose functions with very little training data and minimal data diversity, and we intend to understand what drives the existence of compositionality in autoregressive Transformers trained using the standard training pipeline.
> 2. [1, 2] observe that Transformers and RNNs do not learn to compose on datasets like CTL and CTL++. However, **we present results that, on the contrary**, show that the **standard Transformer** architecture is capable of almost perfect compositional generalization. It would be interesting to understand why these differences exist and this warrants further exploration. We hypothesize that a reliable training pipeline with all the bells and whistles is largely responsible for this difference. No prior work has shown minimal demonstrations, that show this is possible for Transformers that perform stepwise computation–which is very natural in autoregressive models.
> 3. Our **work evaluates functions outside of table lookups** and identifies that compositionality depends on the nature of the composed functions (permutations vs. bijections), which isn’t considered in prior works like [1, 2].
> 4. **The scale of our experiments is larger.** We evaluate these models systematically on as many as 4 million functions (even if the model was trained on as few as 100 of these functions) and take steps towards identifying precise conditions under which compositionality occurs and fails. In comparison some table-lookup tasks like [1, 2] are often limited to evaluation on just 128–256 functions
>
> We also note that we have significantly expanded the Related work section to emphasize some of these points.

---

> ### Author Response · Authors · 2023-11-21
> **Response to Reviewer 1C3m (3/3)**
>
> **>>> finding that step-wise composition is easier than direct is rather unsurprising,**
>
> We arguably agree that this result is unsurprising, but emphasize that the step-by-step prompt format does better with extremely little data: we find that training on as few as 50 functions (1% of total) is enough to perfectly generalize to all other compositions not present in the training data. While prior works on the CTL dataset have identified that Transformers are poor at compositional generalization, our experiments in the step-by-step format highlight why the same need not be true for an autoregressive model, and how it significantly changes such a conclusion.
>
> The compositional abilities of chain-of-thought in language models are still not well understood, For example works like Dziri et al. ([https://arxiv.org/abs/2305.18654](https://arxiv.org/abs/2305.18654)) argue that  language models have limited compositional abilities. Our work is a demonstration that it doesn’t seem to be the case in the synthetic setup even with limited data diversity. We hypothesize that the community may be severely underestimating the capabilities of pretrained language models, since we do not know the ideal prompt (unlike in the synthetic setup) to elicit a capability, even though it may exist in the model.
>
> **>>>  it discusses compositional generalization and the highest most abstract level, at which the answer is: it depends.**
>
> A strong claim requires strong evidence. We disagree that our experiments and discussion is only at the abstract level and we request the reviewer to justify why they think this is the case. Our work demonstrates that Transformers can easily learn to compose under a precisely defined data generating process.
>
> To the best of our knowledge, we are the first to demonstrate how drastically small the training data can be, such that a Transformer learns to compositionally generalize. Our work also characterizes some properties of the training data (in-order compositions, step-by-step format, base21 vs. random) which contribute to compositional generalization. There is little to no other work that precisely characterizes when Transformers learn to compose, let alone demonstrate that it occurs with so little data. Our work makes progress on both these fronts.
>
> **>>>  My understanding is that bijection here means per-token bijection, whereas permutation means shuffling tokens of the word.**
>
> That is indeed correct. We apologize if this wasn’t clear. We have added Figure 10 to Appendix A.1 that hopefully clarifies this detail.
>
> **>>> The details of what the vocabulary size and what the size of the word is were difficult to find**
>
> The details are present in Appendix A. We have also included them in Section 3.2 in the updated version of the manuscript.

---

### Official Review · Reviewer_h95b · 2023-11-01

**Soundness:** 3 good
**Presentation:** 3 good
**Contribution:** 2 fair
**Rating:** 6
**Confidence:** 4

**Summary:**

The authors study how well Transformers can learn to compose functions via a synthetic task of composing bijective functions. They study in-order and out-of-order generalization; and also step-by-step and direct computation the composed functions. They show that Transfomers can generalize well with step-by-step computation but not as well at direct computation. Similarly, in-order compositions are easier to generalize to compared to out-of-order compositions.

**Strengths:**

- The paper is clearly written and easy to follow.
 - The paper introduces a new and useful synthetic task to understand compositional generalization, that will be of use to the research community.
 - The paper illustrates new compositional capabilities and limitations of Transfomers.

**Weaknesses:**

- I think an ablation over the number of layers in the Transformer is a key missing study here. Many studies (e.g. [Weiss et al](https://arxiv.org/abs/2106.06981), [von Oswald et al](https://arxiv.org/abs/2212.07677)) show that the depth of a Transfomer is key to what it can compute, so I'd like to see a sweep over the number of layers in the Transfomer and how that affects compositional generalization.
 - I am not fully convinced that step-by-step computation implies compositional generalization. If the Transfomer is supervised with the outputs of the intermediate function results, isn't it then just learning 1. the individual functions and 2. that it should apply them sequentially? Maybe I'm missing something, but I'd definitely appreciate some sort of discussion around this in the paper.
 - (nitpick, did not affect review score) Suggestion: I think the title of the paper could be a lot better to reflect what's in the paper. Perhaps "Studying compositional generalization in transfomers via synthetic bijective functions" or something along those lines.

**Questions:**

- To test if the Transfomer learned a certain composition of functions, how many examples (i.e. permutations of tokens) are used, and how are they constructed? Are experiments restricted to 6 non-unique tokens per example like in Figure 3? Unless I missed something, I think it'd be useful to clarify this.
 - Will the authors release code/data to reproduce this paper? I imagine the code is not hard to write, but one benefit of such papers that use synthetic tasks and small models is that they should be easy to reproduce.

---

> ### Author Response · Authors · 2023-11-21
> **Response to Reviewer h95b (1/1)**
>
> We thank the reviewer for their time and are glad that they find the paper well-written, the synthetic setup to be useful and our results on compositional capabilities to be new. We hope that the reviewer will champion our paper.
>
> **>>> I’d like to see a sweep over the number of layers in the Transformer and how that affects compositional generalization.**
>
> Thank you for this suggestion! We have two new results that vary the number of layers in a Transformer (Appendix B.1). We find that Transformers **compositionally generalize even with 1 layer and up to 28 layers** in the step-by-step format. Models also compositionally generalizes in the **direct prompt format** but **requires at least 2 to 3 layers** and works only for a composition of a permutation and a bijection.
>
> **>>> isn't it then just learning 1. the individual functions and 2. that it should apply them sequentially?**
>
> We emphasize that the step-by-step prompt format does not necessarily imply compositional generalization and it was not our intent to indicate that this is the case. We have updated the manuscript to address this (for example, see Section 4.1, green text) and have also highlighted two results (one of them new) which show that the ability to compose also depends on the architecture and training data, not just the use of step-by-step prompting format.
>
>
> * On Architecture: We have conducted new experiments (see Appendix B.2)  with **LSTMs and observe that they do not compositionally generalize in the step-by-step and direct prompt formats**. Transformers seem to have an inductive bias to capture compositional structures in the training data. As a result, they can have a significantly larger set of capabilities compared to what’s present in the training data.
> * On training data: The training data also influences if the model learns to compose. Figure 4 shows that training on just the individual functions (21 base, which acts like a null model for the choice of the training data) does not result in compositional generalization. In addition, the model must see compositions of some of these functions in the training data—albeit very few of them—for it to generalize to unseen compositions.
>
> **>>> To test if the Transfomer learned a certain composition of functions, how many examples (i.e. permutations of tokens) are used, and how are they constructed? Are experiments restricted to 6 non-unique tokens per example like in Figure 3?**
>
> These details are present in Appendix A, but to emphasize them further, we have also added them to Section 3.2 (highlighted in Green) in an updated version of the manuscript. In brief: The vocabulary is of size 10 and we sample 6 tokens from this vocabulary to create the input string in all our experiments (like in Figure 3). To test if a Transformer has learnt a certain composition, we compute the accuracy of prompt completions on 1000 samples (10% of the entire input domain).
>
> **>>> (nitpick) I think the title of the paper could be a lot better to reflect what's in the paper.**
>
> Thanks for this suggestion! We will try to select a better title and are currently workshopping a few options. Our choice of current title was motivated by recent works that try to estimate the existence of a capability in a pretrained Language model, or predict its existence in a future version of the models [1,2]. We argue that if a model learns to compose its capabilities, such capability predictions will face challenges and likely underestimate the set of tasks that a pretrained Transformer can learn to perform, i.e., it is difficult to judge “how capable it can become”.
>
> 1. Percy Liang, et al. Holistic evaluation of language models. arXiv preprint arXiv:2211.09110, 2022
>
> 2. Jordan Hoffmann, et al. Training compute-optimal large language models. arXiv preprint arXiv:2203.15556, 202
>
> **>>> Will the authors release code/data to reproduce this paper?**
>
> We commit to releasing all code and data.

---

### Official Review · Reviewer_dYRf · 2023-11-01

**Soundness:** 2 fair
**Presentation:** 2 fair
**Contribution:** 1 poor
**Rating:** 3
**Confidence:** 4

**Summary:**

The research delves into the compositional capabilities of transformers, examining their potential to generalize to functions not present in their training data. Using a synthetic setup, the authors found that transformers can learn and generalize to an exponential or even combinatorial number of functions based on the compositional structure in the data. The nature of the training data plays a pivotal role in this generalization, with step-by-step compositions proving more effective than direct ones. Additionally, attention layers, particularly between layers 6-10, were identified as crucial for compositional generalization. While the study underscores the promise of transformers' compositional abilities, it also highlights the challenges and nuances of using synthetic data and poses further questions about the underlying mechanisms in transformers.

**Strengths:**

None

**Weaknesses:**

It is conceptually wrong to evaluate a  capability of "a Transformer" since it will depend on architecture, training data, training methodology etc.
Moreover I failed to identify which transformer was used in the paper.
Capability of each layer in a transformer is again depends on the number of heads, hidden dimension etc. therefore it is not correct to identify layers 6-10 as crucial layers for compositional generalization.
Overall, I perceive the paper as lacking scientific depth, offering merely a mechanical examination of an ambiguous model without a clear interpretation.

**Questions:**

None

---

> ### Author Response · Authors · 2023-11-21
> **Response to Reviewer dYRf (1/2)**
>
> Thank you for the feedback and for improving the quality of our work. We hope to engage in a discussion and convince the reviewer of the importance of our work. We have made changes to the paper which hopefully addresses the reviewer’s concerns.
>
> **>>> It is conceptually wrong to evaluate a capability of a Transformer since .....**
>
> We would be happy to incorporate suggestions from the reviewer to make the title more precise. Our title draws inspiration from a recent string of works with similar titles (“Transformers learn shortcuts to automata”, “What can transformers learn in-context? a case study of simple function classes”, “On the ability and limitations of transformers to recognize formal languages”). Similar to the motivations of our work, these papers study transformers on synthetic datasets to understand the phenomenology in a controlled setting. Our title also draws inspiration from literature on estimating the “capabilities” of LLMs [1,2,3].
>
> 1. Aarohi Srivastava et al.l. Beyond the imitation game: Quantifying and extrapolating the capabilities of language models. arXiv preprint arXiv:2206.04615, 2022
> 2. Percy Liang, et al. Holistic evaluation of language models. arXiv preprint arXiv:2211.09110, 2022
> 3. Alex Tamkin, et al.. Understanding the capabilities,limitations, and societal impact of large language models. arXiv preprint arXiv:2102.02503, 2021.
>
> **>>> ..... it will depend on architecture, training data, training methodology**
>
> We agree! In fact, we emphasize that our paper’s title is “How capable can a transformer *become*?”; that is, we’re focused on the setting where a Transformer model is trained, via a relatively standard pipeline, on a data domain that matches the compositional nature of language to see if the model can learn to compose its capabilities. While we already rigorously ablate the influence of structure in training data in our experiments, we have added further experiments that we believe, addresses all three aspects raised by the reviewer (architecture, training data, and training methodology).
>
> _Training data:_ As stated in the reviewer’s summary of our work, our work presents experiments that show how “the nature of the training data plays a pivotal role in this generalization”. We precisely characterize how changes to the training data (in-order and out-of-order compositions, base21 and random subsets, step-bv-step and direct prompts) can either result in a combinatorial explosion of capabilities, no compositional generalization or somewhere in between. A surprising observation is that a very small amount of functions in the training data (0.003%  of the total number of functions) is enough to observe compositional generalization when Transformers performs the computations step-by-step.
>
> _Training Methodology:_ We study auto-regressive Transformers trained using the cross-entropy loss. We have made changes to the manuscript to make this more prominent in the introduction and abstract (highlighted in green). We have also added more details to Appendix A.
>
> _Architecture:_ We have added new experiments to Appendix B.1 where we change the number of layers, the number of attention heads, and the embedding dimension. Transformers exhibit compositional compositional generalization (step-by-step and direct formats) even if we vary the number of layers or the number of attention-heads. This ability deteriorates if we reduce the embedding dimension. **This is evidence that our observations are not specific to one single Transformer architecture.**
>
> In addition to these results, we add that many influential previous works, that study Transformers in similarly designed synthetic setups like ours (e.g., [https://arxiv.org/abs/2208.01066](https://arxiv.org/abs/2208.01066)), present experiments on a single Transformer architecture. The tacit assumption in these works is that deep networks of different sizes and embedding dimensions learn similar functions even if they are represented differently using the weights. As a result, studying one architecture translates to other variants of the same architecture.
>
> We also present new experiments (Appendix B.2) that train a set of LSTM on these tasks. **While LSTMs achieve perfect training accuracy, we observe that it fails to compositionally generalize.** This is evidence that auto-regressive Transformers in the step-by-step prompt format have an inductive bias for compositional generalization and all architectures do not work in our synthetic setup.
>
> **>>> Moreover I failed to identify which transformer was used in the paper.**
>
> The details are presented in Appendix A. We use nanoGPT ([https://github.com/karpathy/nanoGPT](https://github.com/karpathy/nanoGPT)) with 12 layers, 12 attention heads and an embedding dimension of size 120.

---

> ### Author Response · Authors · 2023-11-21
> **Response to Reviewer dYRf (2/2)**
>
> **>>>  it is not correct to identify layers 6-10 as crucial layers for compositional generalization**
>
> Thank you for raising this point and we accept we poorly phrased this claim! We intended to say that attention layers in the *latter half* of a Transformer play a crucial layer in compositional generalization. We have modified the claims in the introduction to better address this. We have also run two new experiments to further strengthen our claims empirically.
>
> Linear probing of Transformers of different depths: We conduct  the linear probe analysis on Transformers of different sizes (see Figure 14, Appendix B.4). Across 15 Transformers of different depths, we make two observations: (1) The accuracy increases sharply after an MLP layer for most architectures (2) The composition seems to occur in later layers of a Transformer. We have modified the manuscript to state that: “Our experiments indicate that attention layers in the latter half of a Transformer play a role in compositional generalization”
>
> Attention maps: We have also added Figure 7 (right), where we draw the attention maps for a 1-layer transformer. The map clearly highlights two tokens: the task-token (function to compute) and the intermediate output token to operate on. This experiment could help us build a precise mechanistic description of the role of attention in compositionality. Currently, it provides further credence to our claims on the critical role of attention in compositionality.
>
> **>>>  perceive the paper as lacking scientific depth, offering merely a mechanical examination of an ambiguous model without a clear interpretation.**
>
> We hope to convince the reviewer of the importance of our work. Our goal is not to offer a mechanical examination of compositionality and precisely characterize why it occurs in Transformers. Our goal is demonstrative and we show how Transformers can learn to compose using a small number of functions training data, and we take initial steps to precisely characterize when this occurs.
>
> We restate some our main contributions:
>
>
>
> 1. Our work proposes a **new synthetic setup to systematically study compositional generalization** in autoregressive Transformers
> 2. We show that transformers (with different depths, heads) can **generalize to combinatorially many functions which are entirely out-of-distribution**. The results are surprising: we identify that the step-by-step format with small amounts of diversity in the training data (as few as 0.003% of all the functions) is enough to achieve compositional generalization in Transformers. The ability of a Transformer to output intermediate steps of the computation makes it significantly easier to perform compositions.
> 3. We **precisely characterize conditions on the training data** under which Transformers struggle to compose. We also show that autoregressive LSTMs struggle to compose in the same scenarios where autoregressive Transformers succeed.
> 4. In addition, we offer a preliminary analysis highlighting that attention in the later layers seems critical for compositionality. We also present results that analyze the attention maps of the Transformer which reveal mechanistic insights into how the attention operates.

---

### Author Response · Authors · 2023-11-21
**Common Response to all Reviewers**

We thank the reviewers for their insightful feedback and for taking the time to help us improve the quality of our work. We are glad that the reviewers find our work well-written [h95b, 1C3m], the synthetic setup to be useful for future work [h95b, xno8], the experimental analyses to be new [h95b] and perceptive [xno8].  We have addressed questions from the reviewers in the individual responses below. We have also added new experiments to the paper and have made writing changes that incorporate feedback from the reviewers. The salient changes have been highlighted in green.

We summarize the list of new experiments below:

1. **Attention maps (Figure 7 (right)).** We analyze the attention maps of a 1-layer Transformer for a model trained on the step-by-step prompt. The attention map attends to two tokens: a task token (specifying the function) and an input token to apply the function on. This coupled with the plot in Figure 7(left) suggests that the attention selects the input and function which is computed in the MLP.
2. **Varying the number of layers, embedding dimension and attention heads (Appendix B.1).** We observe that the number of layers and attention heads do not alter our observation on compositionality. However, reducing the embedding dimension causes the accuracy to deteriorate.
3. **Compositionality in LSTMs(Appendix B.2).**  We train LSTMs on data from our synthetic setup and find that it fails to compose functions not present in the training data. Hence Transformers have an inductive bias that helps it compose functions.
4. **Linear probing Transformers with different number of Layers (Appendix B.4).**. We expand on the linear probing results and analyze Transformers with different number of layers and identify that the accuracy increases after the MLP layers across all Transformers
5. **Fixing a bug in Figure 7(left).** This figure on linear probing has slightly changed (probe was tested with incorrect residual stream). This has changed the plots slightly. The plot shows that the attention layers are like inflection points and the accuracy sharply increases after MLP layers in the last few layers of the Transformer.  FIgure 7(left and right) together highlight how attention plays a crucial role in compositionality.

With regards to the writing, we have made the introduction more precise, expanded on the related works section, improved the clarity of notation in Figure 2 and Section 3, improved the description of the experimental setup in Section 3.2, added a discussion to Section 4.1 and expanded the experimental details in Appendix A.

---

### Meta-Review · Area_Chair_B7sW · 2023-12-01

**Metareview:**

This paper studies under which conditions the NanoGPT transformer learns to perform a compositional generalization task, finding that, given appropriate input, the model can successfully generalize from small amounts of data; the training process plays a crucial role; and generation of intermediate outputs is important for fast generalization.

The paper fits into a long tradition of works that have used synthetic tasks to probe neural networks' compositionality abilities. The main novelty is that this work shows that a standard transformer architecture, when appropriately trained, can learn to compositionally generalize in an efficient manner. However, it's not clear what the impact of this finding should be, as it is limited to one particular, small transformer architecture, that is specifically trained to compose (there is a comparison to LSTMs, but the latter are by now obsolete and it is not a novelty that they are unable to compositionally generalize).

Thus, I agree with the general opinion expressed by most reviewers that, while this is a useful addition to the compositionality literature, it might be more appropriate for a targeted workshop than for a general conference such as ICLR.

**Justification For Why Not Higher Score:**

The paper presents an interesting experiment, which however is limited in scope and provides insights whose generality is not clear.

**Justification For Why Not Lower Score:**

NA

---

### Decision · Program_Chairs · 2024-01-16

Reject